# Airqtl dissects cell state-specific causal gene regulatory networks with efficient single-cell eQTL mapping

Matthew W. Funk[1], Yuhe Wang[1,2] & Lingfei Wang [1] ✉

Single-cell expression quantitative trait loci (sceQTL) mapping offers a powerful approach for understanding gene regulation and its heterogeneity across cell types and states. It has profound applications in genetics and genomics, particularly causal gene regulatory network (cGRN) inference to unravel the molecular circuits governing cell identity and function. However, computational scalability remains a critical bottleneck for sceQTL mapping, prohibiting thorough benchmarking and optimization of statistical accuracy. We present airqtl, an efficient method to overcome these challenges through algorithmic advances and efficient implementations of linear mixed models. Airqtl achieves superior time complexity and over $10^8$ times of acceleration, enabling objective method benchmarking and optimization. Airqtl offers de novo inference of robust, experimentally validated cell state-specific cGRNs that reflect perturbation outcomes. Our results dissect the drivers of cGRN heterogeneity and underscore the value of natural genetic variations in primary human cell types for biologically relevant single-cell cGRN inference.

Single-cell technology offers a unique opportunity to infer gene regulatory networks (GRNs) at an unprecedented scale. Recent computational advances[1–4] have enabled the inference of cell type-specific and dynamic GRNs from a single sample. These approaches significantly reduce experimental costs and effort while enhancing the specificity of GRNs for distinct cell types, developmental stages, and extracellular conditions (collectively referred to as "states"). However, many existing methods focus on optimizing predictive performance between genes, often confusing causation with reverse causation or confounding on its own. These noncausal GRNs often fail to predict perturbation outcomes, limiting their reliability in applications such as disease understanding and drug discovery. While some approaches introduced causal evidence with transcription factors (TFs) and their DNA-binding events, these partial GRNs overlook other important regulatory mechanisms, such as signaling pathways, chromatin modifications, and non-coding RNAs.

Causal inference, particularly with perturbations, provides a robust framework for distinguishing causation and accounting for diverse regulatory mechanisms. This paradigm has been successful across various natural and social sciences, including single-cell experiments such as Perturb-seq[5–8]. However, these studies primarily involve engineered cell lines subjected to strong artificial perturbations. Since GRNs are highly dynamic and cell state-specific[1–4], the relevance of these findings to primary cell types in natural human populations with both strong and mild perturbations remains suboptimal. In contrast, Mendelian randomization (MR) leverages natural genetic perturbations as instrumental variables to infer causality between organismal and/ or molecular traits[9–13]. MR has been effectively employed to infer tissue-specific causal GRNs (cGRNs) from population-scale bulk RNA sequencing (RNA-seq), using expression quantitative trait loci (eQTLs) that directly regulate cis-gene expression and indirectly influence trans-genes[14].

Applying MR to population-scale single-cell RNA-seq (scRNA-seq) promises greater cell state specificity and cost-effectiveness than bulk RNA-seq for cGRN inference. While dozens of population-scale scRNA-seq studies have been conducted (e.g.,[15–19]), the

[1]Department of Genomics and Computational Biology, UMass Chan Medical School, Worcester, MA, USA. [2]Department of Computer Science, Metropolitan College, Boston University, Boston, MA, USA. ✉e-mail: Lingfei.Wang@umassmed.edu

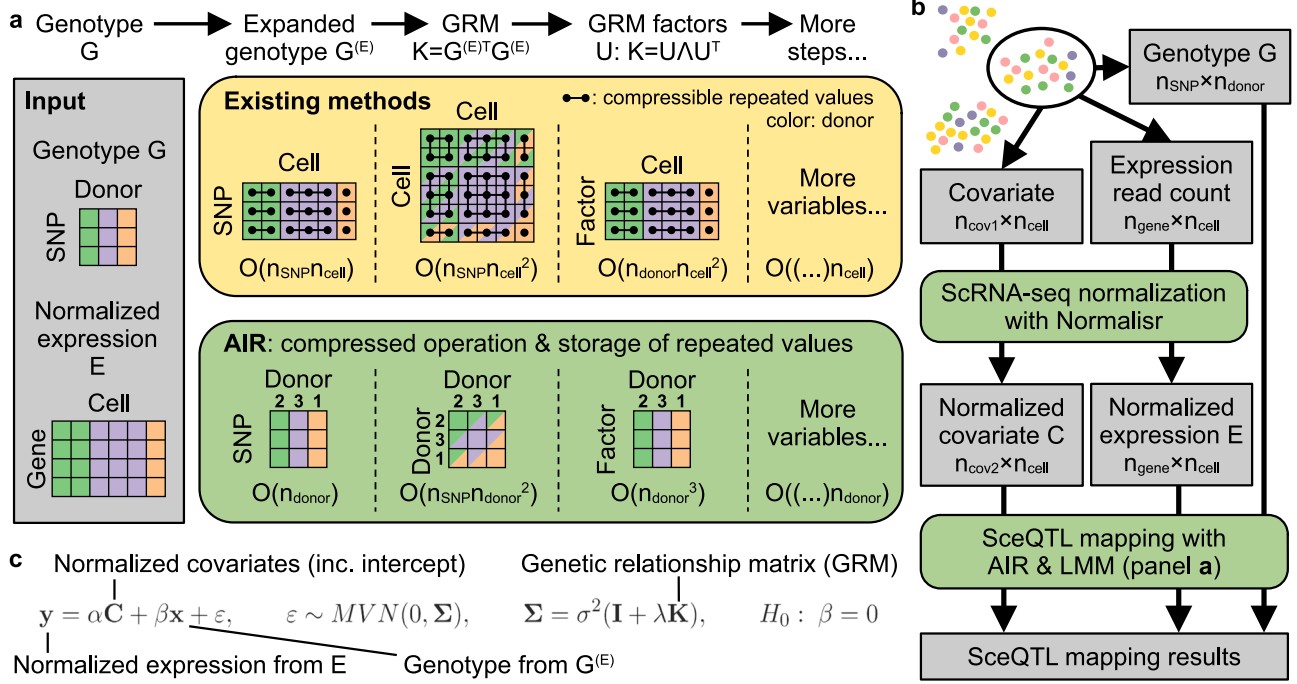

**Fig. 1 | Overview of algorithmic acceleration from AIR and sceQTL mapping with airqtl. a** AIR acceleration in key LMM steps. Existing methods expand the genotype matrix from donor-level (color) to cell-level, creating repeated rows and columns (dots connected by edges) that incur computational inefficiency. AIR instead operates on the compressed matrix and the number of repeats (**2**, **3**, **1** shown), enabling mathematically equivalent but more efficient computations. The time complexity for each method (box color) is indicated below each step. **b** Workflow of sceQTL mapping with airqtl for each selected cell type using only data from the corresponding cells. **c** Linear mixed model and null hypothesis tested by airqtl for sceQTL mapping between each genotype and gene expression level. Box color (**a**, **b**): method (yellow: existing; green: AIR/airqtl; gray: data). Matrix color (**a**) and schematic low-dimensional embedding color (**b**): donor. Each entry in the GRM **K** (**a**) can be shaded in up to two colors because its row and column indices can be associated with two different donors.

challenges in single-cell eQTL (sceQTL) mapping[15,20] have limited its utility for downstream cGRN inference. On the statistical front, single-cell data sparsity necessitates specialized normalization techniques[5,21,22], while the shared genome of cells from the same donor requires (generalized) linear mixed models[16,23]. However, the primary bottleneck lies in computational scalability. Without scalable methods, it is impractical to benchmark and improve statistical performance at a reasonable cost and time. Consequently, most population-scale scRNA-seq studies rely on ad hoc sceQTL mapping computer programs, with few capable of mapping all cis- and trans-sceQTLs across all single nucleotide polymorphism (SNP)-gene pairs. Some studies employ pseudo-bulk strategies for eQTL mapping, but they face challenges with continuous cell states, technical confounders, and limited statistical power[15,20,24]. Accurate and efficient eQTL mapping is also pivotal in human genetics[25] and genomic foundation models[26,27]. The absence of a scalable sceQTL mapping method has hindered progress in these domains.

In this paper, we present airqtl, an efficient method to overcome these challenges in sceQTL mapping and infer cell state-specific cGRNs. Airqtl introduces a scalable exact algorithm that accelerates basic matrix operations by leveraging the donor-cell hierarchy in population-scale single-cell datasets. This advancement improves the time complexity of linear mixed models beyond state-of-the-art methods such as FaST-LMM[28] and GEMMA[29]. Together with GPU utilization and other implementation optimizations, airqtl achieves over eight orders of magnitude in computational acceleration at a comparable or higher statistical accuracy. This scalability enables objective benchmarking and refinement of cell state-specific sceQTL mapping and admixed population modeling. Finally, airqtl's efficient and accurate sceQTL mapping allows the de novo inference of robust, experimentally validated cell state-specific cGRNs. Comparative analyses reveal intrinsic and extrinsic drivers of

cGRN heterogeneity, underlining the importance of cell state-specific experimental data and computational methods.

## Results

### Airqtl overview

The core of airqtl is an Array of Interleaved Repeats (AIR), a compressed data structure designed to efficiently handle repeated values, a pattern common in sceQTL mapping (Fig. 1a). SceQTL mapping requires genotype matrix expansion from donor- to cell-level, leading to repeated values that incur substantial computational and storage overhead. AIR circumvents this inefficiency by storing only their original values and repeat patterns, enabling scalable exact algorithms to accelerate basic matrix operations essential for sceQTL mapping, such as multiplication and singular value decomposition (Supplementary Note).

For instance, AIR factorizes cell-level genetic relationship matrix (GRM) in $\mathcal{O}(n_{donor}^3)$ time complexity, compared to $\mathcal{O}(n_{donor}n_{cell}^2)$ required by state-of-the-art methods like FaST-LMM[28] and GEMMA[29]. This principle underpins similar accelerations across all linear mixed model (LMM) steps in sceQTL mapping, yielding a total time complexity of $\mathcal{O}(n_{gene}n_{cell} + (n_{SNP} + n_{donor})(n_{gene} + n_{donor})n_{donor})$ (**Methods,** Supplementary Note). When compared against the current state-of-the-art time complexity[28,29] of $\mathcal{O}((n_{SNP} + n_{donor})(n_{gene} + n_{cell})n_{cell})$, AIR's algorithm alone would deliver a ~1000-fold speedup for sceQTL mapping (**Methods**).

Airqtl performs sceQTL mapping for each cell state separately using a streamlined two-step process (Fig. 1b, **Methods**). First, quality-controlled scRNA-seq read counts and covariates are normalized for each cell state using Normalisr[5]. Normalisr is optimized for scRNA-seq normalization and differential expression analysis with linear models, delivering both high accuracy and exceptional speed compared to generalized linear

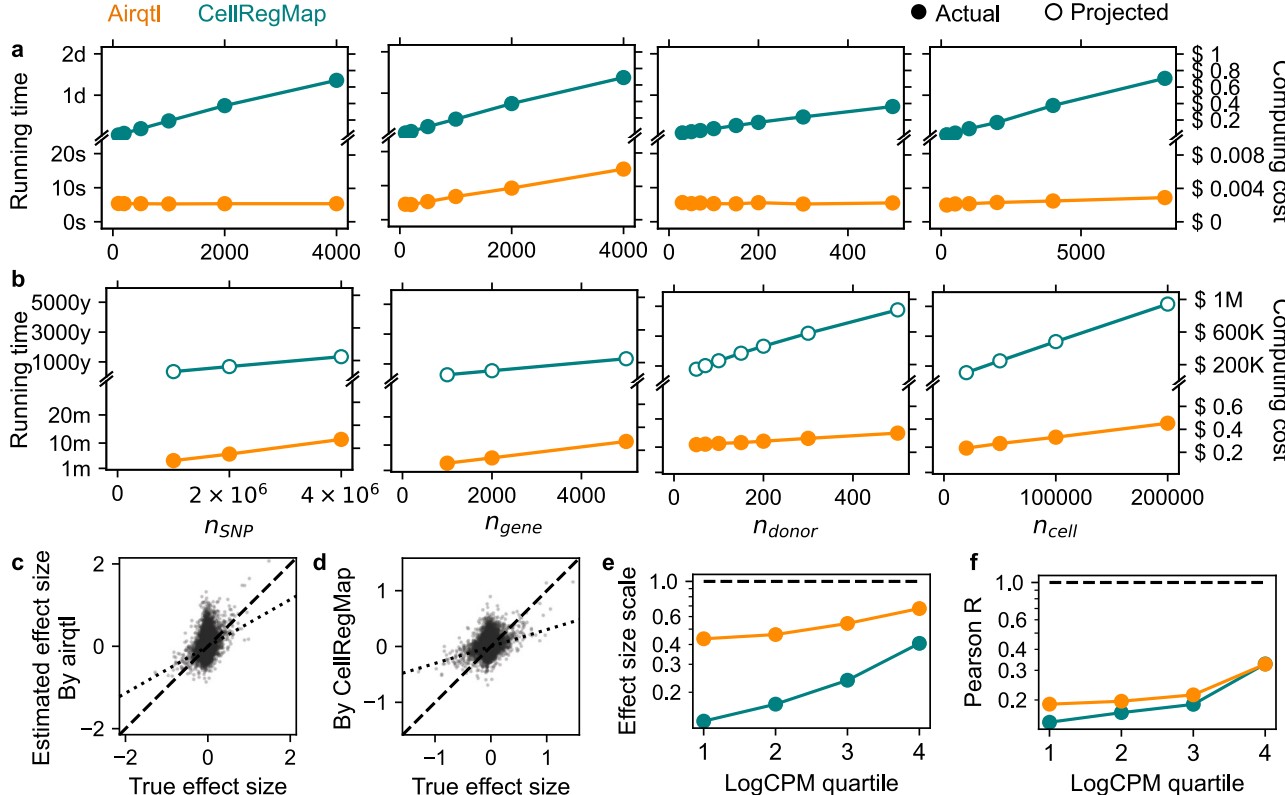

**Fig. 2 | Airqtl offers eight orders of magnitude of acceleration and improved effect size estimation in sceQTL mapping. a**, **b** Running time (left Y) and computing cost (right Y) for sceQTL mapping on small-scale (**a**, defaults to $n_{SNP} = 500$, $n_{gene} = 500$, $n_{donor} = 100$, $n_{cell} = 1,000$) and realistic large-scale (**b**) defaults to $n_{SNP} = 4,000,000$, $n_{gene} = 5,000$, $n_{donor} = 100$, $n_{cell} = 50,000$) datasets with varying dimension sizes (X). **c**, **d** Ground-truth (X) and estimated (Y) sceQTL effect sizes

(dots) by airqtl (**c**) and CellRegMap (**d**) without stratification. Deviation of the best-fit linear model (dotted line) from the diagonal (dashed line) indicates the extent of overall effect size underestimation. **e**, **f** Effect size estimation bias (**e**) and variance (**f**) for genes across expression quartiles (X). Dashed line: perfect performance. Color: method.

model-based methods. Normalisr's accuracy and efficiency has been uniquely demonstrated in mapping the cis- and trans-effects of CRISPRi gRNAs and inferring cGRNs from large-scale Perturb-seq studies with high multiplicity of infection, a scenario that closely resembles sceQTL mapping.

Next, airqtl applies LMMs to identify sceQTLs from the normalized data (Fig. 1c, **Methods**). LMMs are particularly suited for scRNA-seq data as it accounts for the genetic similarity among cells from the same donor using the genetic relationship matrix (GRM). This approach is analogous to its use in genome-wide association studies (GWAS) for handling population structures, such as twins, to improve statistical power and reduce false positives[30]. Besides the algorithmic accelerations provided by AIR (Fig. 1a), airqtl implements additional optimizations such as GPU utilization and out-of-core computing. Airqtl also efficiently maps cell type/state-specific sceQTLs with an extended LMM including all linear and quadratic interaction terms that receives similar accelerations (**Methods**).

**Efficient and accurate sceQTL mapping with airqtl**

We conducted comprehensive benchmarking to evaluate the performance improvements offered by airqtl. In GWAS and eQTL mapping, simulated datasets have been pivotal for method benchmarking due to the lack of ground-truth information[28,31–33]. Following this precedent, we simulated population-scale scRNA-seq datasets of various sizes, incorporating ground-truth characteristics closely resembling real-world studies (**Methods**). These simulations were based on Randolph et al.[19], a published scRNA-seq dataset comprising influenza A virus (flu) or mock (NI)-exposed human peripheral blood mononuclear cells

(PBMCs) from male donors of African and European descent. This dataset was subsequently used for sceQTL mapping and cGRN inference in this study.

We benchmarked airqtl against CellRegMap[23], the only existing method explicitly designed for sceQTL mapping with frequentist inference. We excluded other methods such as GASPACHO[34] and SAIGE-QTL[24] for three reasons: i) their Bayesian or heuristic frameworks reduce compatibility with downstream analyses or population-scale scRNA-seq datasets (**Methods**); ii) they lack formal statistical tests for sceQTL context specificity; iii) we encountered critical challenges while trying to run these methods on official or custom datasets.

Due to CellRegMap's inability to scale to full cis- and trans-sceQTL mapping in large population-scale scRNA-seq datasets, we first tested both methods on small-scale datasets (Fig. 2a), where airqtl already demonstrated substantial advantage. Because running time often scales nonlinearly with problem dimensions, we employed a log-log model to extrapolate CellRegMap's running time for large-scale datasets and compared it with the actual runtime of airqtl (**Methods**). The results revealed that airqtl achieved over eight orders of magnitude of acceleration and six orders of magnitude of cost reduction (Fig. 2b).

Accurate effect size estimation is crucial for eQTL mapping and downstream analyses, such as cGRN inference. However, it is often overlooked in single-cell differential expression studies and is particularly challenging due to expression-dependent biases[5]. To assess this issue in sceQTL mapping, we followed a bias-variance decomposition approach[5], stratifying errors by expression level (**Methods**). Airqtl greatly mitigated the expression-dependent underestimation bias of sceQTL effect sizes observed in CellRegMap (Fig. 2c–f). Additional analyses attributed this improvement to the use of Normalisr for

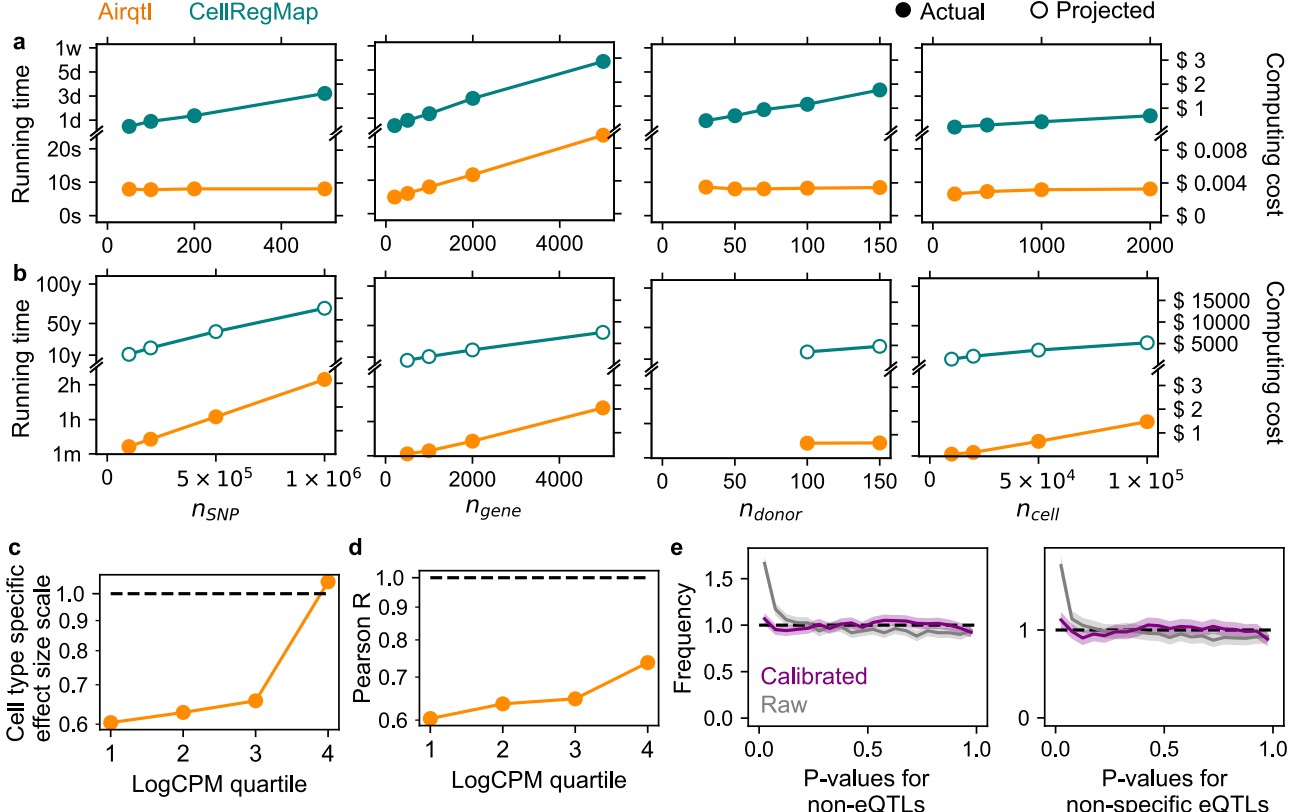

**Fig. 3 | Airqtl enables efficient cell type-specific sceQTL mapping, comprehensive benchmarking, and objective optimization. a, b** Running time (left Y) and computing cost (right Y) for cell type-specific sceQTL mapping on small-scale (**a**, defaults to $n_{SNP} = 200$, $n_{gene} = 1,000$, $n_{donor} = 50$, $n_{cell} = 2,000$) and large-scale (**b** defaults to $n_{SNP} = 200,000$, $n_{gene} = 2,000$, $n_{donor} = 100$, $n_{cell} = 50,000$) datasets with varying dimensions. Color: method. **c, d** Low bias (**c**) and variance (**d**) in cell type-specific sceQTL effect size estimation by airqtl, stratified by expression quartiles (X). Dashed line: perfect performance. **e** Airqtl's scalability enabled benchmarking and optimization of $P$ value calibration for cell type specificity. Null $P$ value histograms are shown for non-eQTLs (left) and non-specific eQTLs (right) before (raw) and after (calibrated) calibration. Shades in histograms represent error bars ($3\sqrt{N}$, where $N$ is the number of entries in each bin). Dashed line: perfect performance (standard uniform distribution).

scRNA-seq normalization (Supplementary Fig. 1, and Supplementary Note), which has been shown to reduce such biases in differential expression analyses[5]. This advantage is expected and confirmed in Supplementary Note to be especially critical for cell type/state-specific sceQTL mapping, where expression-dependent biases can erroneously attribute changes in expression levels as cell type/state-specific sceQTLs.

Besides its advancements in efficiency and effect size estimation, airqtl demonstrated comparable or superior sensitivity and specificity (Supplementary Fig. 2, and Supplementary Data 1, **Methods**). Taken together, airqtl provides a robust and comprehensive solution for efficient and accurate sceQTL mapping.

### Efficient and optimized cell type-specific sceQTL mapping

The interaction between genotype and other factors is well established in genetics[35,36]. Numerous bulk and single-cell eQTL studies have revealed tissue or cell type-specific eQTL effects[15–19,37]. While sceQTLs have been mapped separately for each cell type, statistical tests are essential to differentiate true cell type specificity from randomness in data but remain rare and underexplored. Airqtl overcomes these limitations by enabling efficient cell type-specific sceQTL mapping with an extended linear mixed model that incorporates linear and quadratic interactions between genotype, cell type, and other covariates (**Methods**). To systematically evaluate performance, we simulated population-scale scRNA-seq datasets with cell type-specific eQTLs based on CD4+ and CD8+ T cell subpopulations from the Randolph et al. dataset[19] (**Methods**).

In terms of computational efficiency, airqtl was over five orders of magnitude faster and three orders of magnitude cheaper than Cell-RegMap (Fig. 3a, b, **Methods**). This scalability enables genome-wide mapping of cell type-specific sceQTLs. Although CellRegMap's limited scalability prevented statistical benchmarking on large datasets, airqtl showed robust performance in estimating cell type-specific effect sizes (Fig. 3c, d, and Supplementary Fig. 3, Supplementary Fig. 4, **Methods**). Interestingly, cell type-specific effects were easier to estimate than sceQTL effects (Fig. 2ef), displaying lower bias and variance.

False positives in cell type specificity, in the form of biased null $P$ value distributions, can lead to unnecessary and costly follow-up experiments. Initial results from airqtl showed an overabundance of small null $P$ values for both non-eQTLs and non-specific eQTLs (Kolmogorov-Smirnov/KS $P < 10^{-99}$ and $P < 10^{-50}$ respectively, Fig. 3e, and Supplementary Data 2, **Methods**). Given the weak but nonzero expression-dependent biases in sceQTL effect size estimation (Fig. 2e), this result was anticipated and motivated an optimization of the model. To address this, we calibrated $P$ values using Beta distributions following existing bulk eQTL studies[37,38]. To overcome expression-dependent biases in scRNA-seq, we further incorporated stratification with additional parameters. We evaluated 10 candidate parameters, including known single-cell differential expression confounders, their derived parameters, and key population genetics parameters (**Methods**). Each parameter was used individually to parameterize the Beta distribution, which then models the null $P$ values obtained from separately simulated datasets (Supplementary Fig. 5). The learned parameterization was then used to calibrate $P$ values in the original

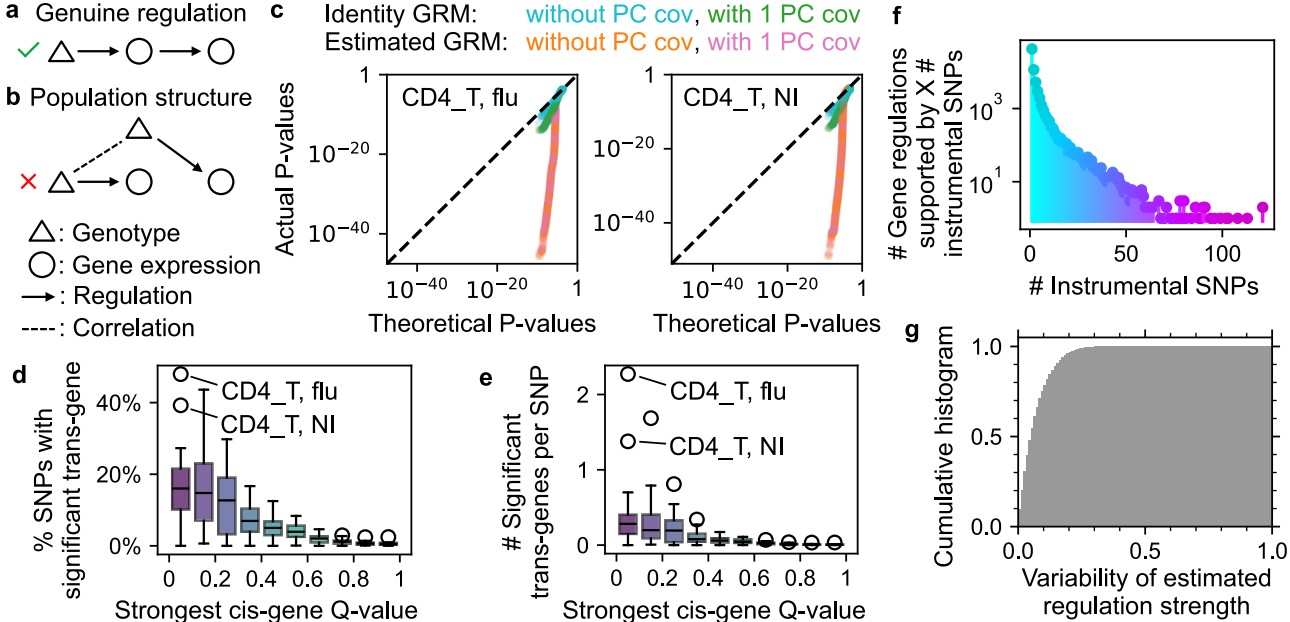

**Fig. 4 | Airqtl enables efficient inference of cGRNs through Mendelian randomization. a, b** Scenarios for true positive (**a**) and false positive (**b**) cGRN inference using genotypic variation as an instrumental variable. Population structure (**b**) can lead to false positives in trans-eQTL mapping and cGRN inference. **c** The vanilla method (identity GRM without PC covariate) produced the best quantile-quantile (QQ) plots of null trans-sceQTL *P* values for different cell states (panel) among different strategies to handle admixed populations (color). Dashed line: perfect performance. **d, e** Airqtl identified abundant trans-genes (Y) for cis-sceQTL SNPs, binned by the strongest cis-sceQTL *Q* value (X). Data point: cell state at each bin. **f** Numerous inferred gene regulations (Y) were supported by multiple unique instrumental SNPs (X) in each cell state. **g** Gene regulation strength estimates were consistent across different instrumental SNPs.

dataset. Calibration using the parameter "n0min" (proportion of zero-read cells for one gene, minimum across cell types compared) yielded the best null *P* value performance, with KS *P*=0.009 and *P*=0.12 for non-eQTLs and non-specific eQTLs, respectively (Supplementary Data 2, Supplementary Fig. 6). This optimization substantially improved the uniformity of null *P* value distributions (Fig. 3e) and also the separate metric of Storey's $\pi_1$ (raw: 0.08, calibrated: 0.006)[39].

In summary, airqtl provides an efficient and scalable solution for cell type-specific sceQTL mapping, uniquely enabling comprehensive benchmarking and objective optimization of statistical performance.

**Efficient sceQTL mapping enabled cGRN inference**

Genetic variations are effective instrumental variables for inferring cGRNs via Mendelian randomization (MR), given their predominantly cis-regulatory effects. By identifying instrumental (variable) SNPs and their significant cis- and trans-genes, causal regulations from cis- to trans-genes can be inferred (Fig. 4a). Population-scale scRNA-seq studies offer unprecedented power for cell state-specific cGRN inference but also pose substantial scalability challenges. Airqtl's unique efficiency provides a solution.

Population admixture is a well-known source of false positives in GWAS[40], yet its impact on eQTL mapping and cGRN inference remains less understood (Fig. 4b). The Randolph et al. dataset[19] includes admixed populations with strong inter-chromosomal genotype correlations (Supplementary Fig. 7). While principal component (PC) covariates of the genotype matrix can capture much of this correlation (Supplementary Fig. 7), they may also introduce spurious associations via collider bias[41]. An alternative approach involves estimating a genetic relationship matrix (GRM) from genotype data[42]. To assess these methods, we compared two donor-level GRMs (identity matrix versus genotype-based GRM) with and without PC covariates. Using SNPs with highly insignificant cis-associations (all raw *P* > 0.5) as negative controls for trans-sceQTL mapping on different chromosomes (**Methods**), the vanilla approach (identity GRM without PC covariates) outperformed other methods in 7/12 cell states

(cell type-flu exposure combination, Fig. 4c, and Supplementary Fig. 8). Notably, genotype-based GRMs had almost identical performances with/without PCs in the QQ plots. This indicated little impact from adding PC covariates, possibly because GRMs already effectively account for admixed population structure. Airqtl's speed enabled mapping of the trans-effects of insignificant SNPs, warranting further investigations into population admixture modeling.

Therefore, we used the vanilla model to map sceQTLs for each cell state (Supplementary Data 3). Strong cis-sceQTL SNPs (BH Q < 0.1) were more likely to exhibit trans-associations and had more significant trans-genes (Fig. 4d, e, BH Q < 0.5), supporting their utility for cell state-specific cGRN inference. To minimize pleiotropy-induced false positives, possible pleiotropic SNPs that may affect multiple cis-genes were excluded (**Methods**). Each remaining SNP, namely "instrumental SNP", could then infer causal gene regulations from its cis-gene to associated trans-genes via two-stage least squares (2SLS, **Methods**).

Multiple instrumental SNPs allow to evaluate and improve the robustness of inferred cGRNs. Combining gene regulatory pairs (namely gene regulations) identified for each cell state, we identified 28,171 gene regulations supported by ≥2 unique instrumental SNPs (Fig. 4f). Among these, only 1 (0.004%) displayed conflicting directionality predictions (activation vs. repression, **Methods**). Additionally, gene regulation strength estimates were highly consistent across different instrumental SNPs, with 75% exhibiting very low relative variability (< 10%, Fig. 4g, **Methods**). These findings reaffirm the robustness of cGRNs inferred from population-scale scRNA-seq data. Our final cell state-specific cGRNs consisted of gene regulations supported by at least two unique instrumental SNPs, whose regulation strength estimates were aggregated with median (Supplementary Data 5).

**Intrinsic and extrinsic drivers of cGRN state specificity**

Single-cell cGRNs offer considerable advantages over noncausal GRNs for understanding molecular circuit rewiring, as they directly quantify perturbation outcomes and indirectly link to organismal trait

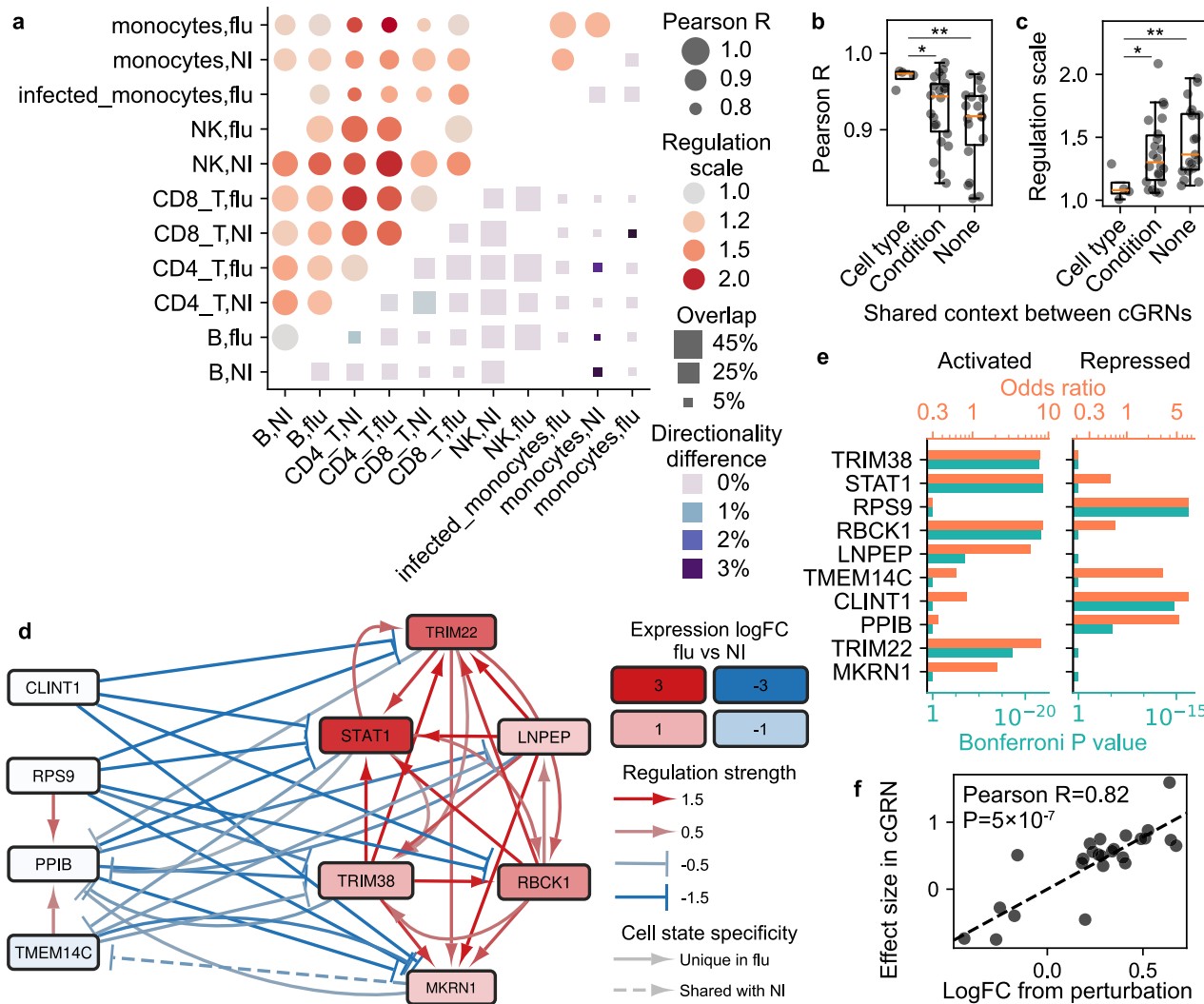

**Fig. 5 | Comparative analyses revealed intrinsic and extrinsic drivers of cGRN cell state specificity. a** State-specific cGRNs showed moderate overlap but high agreement among the overlapped gene regulations in pairwise comparisons. **bc** Pearson R (**b**) and regulation scale (**c**) comparisons across cGRNs with varying shared contexts (X) highlighted cell type as a stronger determinant of cGRN similarity than condition. Identical cGRNs result in Y=1 in both panels. Two-sided Mann-Whitney U test *P*: < 0.1 (*); < 0.01 (**) (Supplementary Data 4). **d** Sub-cGRN between top master regulators was predominantly specific to CD4+ T cells under the flu condition. **e** Enrichment odds ratio (top X) and two-sided hypergeometric *P* value (bottom X) of "defense response to virus" among activated (left) and repressed (right) target genes for each master regulator (Y). **f** Validation of inferred cGRNs using published *STAT1* perturbation experiment in CD4+ T cells. The estimated effect sizes of *STAT1* regulating other genes (dot) from the cGRN (Y) showed high consistency with estimations from the perturbation experiment (X). Dashed line: best linear fit with two-sided Pearson test.

outcomes. To delineate distinct modes of cGRN rewiring—including detectability, directionality, and strength changes—we first performed pairwise comparisons across cell states (Fig. 5a, **Methods**). Detectability varied widely between cell states, reflected in differing overlap rates. These variations stem from both genuine (in)activation of specific gene regulations and technical factors such as cell count. Among overlapping regulations, directionality was highly consistent, with differences confined to a small fraction (up to 3%), which could arise from chromatin-level regulation rewiring. We further quantified changes in regulation strength with regulation scale and Pearson R, reflecting differences in overall magnitude and individual gene regulations, respectively, akin to our benchmarks for sceQTL effect size estimation (Fig. 2e, f and Fig. 3c, d, **Methods**). Pearson R consistently exceeded 0.8 between all cell states, though regulation scale exhibited up to a twofold variation (Fig. 5a). Based on these metrics, cGRNs from the same cell type but different conditions were most similar, followed by those from identical conditions (Fig. 5a–c, Supplementary Data 4). These findings suggest limited cGRN rewiring in regulation strength

and directionality, driven predominantly by cell type and followed by external condition.

To dissect genuine (in)activation of specific gene regulations from technical detectability variations, we next compared CD4+ T cells under flu vs. NI conditions, the most abundant cell states in the dataset with the highest numbers of trans-eQTLs (Fig. 4d, e). Under the flu condition, we identified 10,195 regulations involving 1,938 genes (Supplementary Fig. 9, Supplementary Data 5), of which 8,539 (84%) were cell state-specific and absent in NI. Notably, most regulators (641/681 or 94%) were not known TFs (based on known motifs[4], **Methods**), highlighting airqtl's unique capacity for de novo inference of cell state-specific cGRNs. A force-directed layout segregated genes in the cGRN into two distinct clusters that repressed each other (**Methods**), with upregulated genes enriched in one cluster (Supplementary Fig. 9).

We further focused on the top 10 master regulators with the highest number of target genes. These master regulators segregated into two distinct groups, each connected exclusively by activation edges within the group and repression edges between groups

(Fig. 5d). Notably, 50/51 (98%) regulations among these master regulators were specific to the flu condition. In contrast, the top 10 master regulators under the NI condition exhibited only 21/89 (24%) regulations as cell state-specific (Supplementary Fig. 10). Furthermore, 5/6 master regulators within the upregulated group (TRIM38, STAT1, RBCK1, LNPEP, and TRIM22) had no detectable target under the NI condition. Since all master regulators and many target genes remain expressed in both conditions (Supplementary Data 5), the cell state specificity is consistent with changes in gene regulation (cGRN edge) instead of (in)activation of individual genes (cGRN node). A plausible explanation is that the NI condition predominantly maintains house-keeping regulations while lacking the key gene regulations activated under the flu condition. These findings underscore that active cGRNs are highly specific to the cell state, including both cell type and condition, thereby necessitating cell state-matched data for accurate characterization.

Most of the identified top master regulators[43–53] and their target genes are well-established components of cellular response to viral infections. The activation or repression targets of most (7/10) master regulators were most significantly enriched for the gene ontology (GO) category "defense response to virus" (Bonferroni hypergeometric $P$ values $< 0.05$, Fig. 5e, Supplementary Data 6, **Methods**). Importantly, the activation/repression target gene set exhibiting this enrichment aligned with the regulatory groups of each master regulator shown in Fig. 5d. In contrast, target genes regulated in the opposite direction showed no statistically significant enrichment in any GO category. These enrichments were driven by diverse target genes, with ~20% unique to a single master regulator (Supplementary Fig. 11). These findings confirm that the inferred cGRNs and their directional regulation information are biologically meaningful and highly specific to the cell state.

Finally, we validated the cell state-specific cGRNs using published perturbation experiments in CD4+ T cells. We first queried the whole Gene Expression Omnibus (GEO) for datasets targeting any master regulator and identified one dataset[54] targeting STAT1, which also happened to match the viral exposure cell state. Among their overlapping targets, the cGRN's estimated effect sizes for STAT1 were highly consistent with the perturbation experiment, with Pearson R=0.82 and $P = 5 \times 10^{-7}$ (Fig. 5f). We further found one STAT1 Chromatin Immunoprecipitation sequencing (ChIP-seq) dataset in human CD4+ T cells in the whole GEO[55]. Using this dataset, we identified STAT1's putative direct targets through active binding sites (**Methods**). These exhibited substantial overlap with STAT1 targets in our cGRN (Supplementary Fig. 12, Supplementary Fig. 13, Supplementary Data 7), while the partial overlap confirmed airqtl's ability to capture indirect targets undetectable by ChIP. Together, these results highlight the capability of our method to infer cell state-specific cGRNs that are experimentally validated and directly reflect perturbation outcomes.

## Discussion

We developed airqtl, an efficient and accurate method for mapping sceQTLs and inferring cGRNs from population-scale scRNA-seq data. By integrating computational advances in algorithm design and implementation with established linear mixed models, airqtl achieves a substantial reduction in computational complexity compared to state-of-the-art methods. This yields orders-of-magnitude faster sceQTL and cell type-specific sceQTL mapping, all while maintaining comparable or superior accuracy. Its unprecedented efficiency enables comprehensive benchmarking and objective optimization of cell type-specific sceQTL mapping and population admixture handling. By efficiently mapping all cis- and trans-sceQTLs, airqtl uniquely infers de novo cell state-specific cGRNs that reflect perturbation outcomes in primary human cell types. The inferred cGRNs demonstrated consistency with established biological knowledge and were validated against published perturbation experiments.

Although we demonstrated airqtl's robust statistical accuracy in sceQTL mapping and introduced several optimizations, this study is not a systematic benchmarking or optimization of statistical performance. Instead, the orders of magnitude of acceleration offered by AIR overcame the scalability barrier of these questions and enabled their dedicated future studies. A promising future direction would be representing cell state-specific linear cGRNs as a unified nonlinear cGRN operating on diverse expression states. Our analyses also highlighted several key challenges to be further investigated in (cell type-specific) sceQTL mapping, including effect size estimation, biased null $P$ values, and the complexities of analyzing admixed populations. Beyond airqtl, AIR can similarly empower other existing or future sceQTL mapping methods, as well as dynamic sceQTL or other molecular QTL mapping tasks that follow similar hierarchical structures. Consequently, the value of AIR acceleration remains significant, even if future methods achieve superior statistical performance.

We established the feasibility of inferring cell state-specific cGRNs from population-scale scRNA-seq datasets, paving the way for further benchmarking and improvement of accuracy in future studies. Unlike GRNs based on DNA-binding or expression predictivity that struggle with perturbation outcome prediction[56], causal inference methods are specifically designed for this challenge with enormous successes across disciplines. This is evidenced by our accurate prediction of STAT1 perturbation effects (Fig. 5f) and previous successes with Normalisr[5]. Our findings highlight the intrinsic and extrinsic drivers of cGRN specificity and underscore the necessity of cell state-matched data for deciphering cell state-specific gene regulations. The high level of cell state specificity we observed, consistent with other studies[1–4], underscores both the biological importance of context-specific cGRN inference and the unique value of population-scale scRNA-seq for studying primary human cells under natural genetic variations—advantages unattainable through bulk sequencing or Perturb-seq experiments in engineered cell lines under strong loss-of-function perturbations.

## Methods
### Airqtl
**Array of Interleaved Repeats (AIR).** AIR is a compressed data structure optimized for matrices containing repeated rows or columns, a common pattern in hierarchical data structures. Rather than fully expanding matrices, AIR maintains: i) the compressed matrix containing only one copy of each repeated rows/columns and ii) the number of repeats. This design enables faster matrix operations (addition, multiplication, singular value decomposition or SVD) by operating directly on compressed matrices. The acceleration leverages several mathematical properties including computing compressed results, variance rescaling, and other operand preprocessing. Based on and compatible with torch.Tensor, AIR executes all matrix operations within pytorch, simultaneously benefiting from AIR's scalable algorithms and pytorch's efficient GPU acceleration.

See Supplementary Note for algorithmic details. See "SceQTL mapping—AIR accelerations" in **Methods** below for AIR's application to sceQTL mapping acceleration.

**SceQTL mapping—linear mixed model.** For sceQTL mapping, we employed the LMM described in Fig. 1c using restricted maximum likelihood (reML)[28,29]. Briefly, for each gene, $\lambda$ was first estimated as $\hat{\lambda}$ via maximum likelihood under the null hypothesis ($\beta = 0$ for all SNPs). Genes with over 75% of unexplained variance attributable to donor-level variation (i.e., $\hat{\lambda} > 3$) were excluded from downstream analyses, as such high donor-level variation relative to scRNA-seq measurement noise likely indicates strong technical batch effects. Then, for each SNP, the reML was computed by fixing $\lambda = \hat{\lambda}$ and optimizing all other parameters ($\alpha, \beta, \sigma$) to maximize the likelihood. The parameter estimates were obtained through reML. Hypothesis testing was conducted

using the likelihood ratio test (LRT) to compare the full model against the null model under the same restriction $\lambda = \hat{\lambda}$. In practice, we equivalently tested conditional $R^2$ (coefficient of determination) to obtain $P$ values after transforming the data with the GRM and removing covariates. Under the null hypothesis, the conditional $R^2$ follows a beta distribution, which can be well-approximated by a $\chi^2$ distribution for computational efficiency (default in airqtl).

**Cell state specific sceQTL mapping—linear mixed model.** We used the following model to map the cell state specificity of each sceQTL effect size:

$$\mathbf{y} = \alpha_0 + \alpha_1(\mathbf{C}, \mathbf{s}, \mathbf{x}) + \alpha_2(\mathbf{C}, \mathbf{s}) \otimes (\mathbf{C}, \mathbf{s}) + \alpha_3 \mathbf{C} \otimes \mathbf{x} + \alpha_4 \mathbf{x} \otimes \mathbf{x} + \beta \mathbf{s} \otimes \mathbf{x} + \varepsilon. \tag{1}$$

This model incorporates all linear and quadratic interaction terms between genotype, cell state, and other covariates. Here, $\mathbf{x} \in \{0,1,2\}^{n_{cell}}$ and $\mathbf{y} \in \mathbb{R}^{n_{cell}}$ denote bi-allelic genotype and normalized expression level data for the SNP and gene considered, subjected to the same QC and normalization steps as in sceQTL mapping. $\mathbf{C} \in \mathbb{R}^{n_{covariate} \times n_{cell}}$ and $\mathbf{s} \in \{0,1\}^{n_{cell}}$ include all covariates, with $\mathbf{s}$ specifically containing the cell state covariate (e.g., cell type) being tested for eQTL specificity. $\alpha_0$ is the intercept, and parentheses indicate row-wise matrix concatenation. Covariates $\mathbf{C}$ include scRNA-seq technical covariates from Normalisr[5] to capture library size confounding and any other data-specific or user supplied covariates (e.g., see "Latent dimensions as covariates" below). Coefficients $\alpha_0 \in \mathbb{R}, \alpha_1 \in \mathbb{R}^{n_{covariate}+2}, \alpha_2 \in \mathbb{R}^{(n_{covariate}+1)(n_{covariate}+2)/2}, \alpha_3 \in \mathbb{R}^{n_{covariate}}, \alpha_4 \in \mathbb{R}$ are effect sizes from covariates and $\beta \in \mathbb{R}$ is the effect size from the tested term of cell state specific sceQTL.

Here we defined the *modified row-wise Kronecker product* $\otimes$ satisfying the following conditions. i) For $\mathbf{A} \in \mathbb{R}^{n_A \times k}$ and $\mathbf{B} \in \mathbb{R}^{n_B \times k}$, $\mathbf{A} \otimes \mathbf{B} \in \mathbb{R}^{n_A n_B \times k}$ computes the element-wise product between each row pair of $\mathbf{A}$ and $\mathbf{B}$. ii) Vectors are treated as 1-row matrices: $\mathbf{A} \in \mathbb{R}^{n_A \times k}$ and $\mathbf{x} \in \mathbb{R}^k$, $\mathbf{A} \otimes \mathbf{x} \in \mathbb{R}^{n_A \times k}$ regards $\mathbf{x}$ as a matrix in $\mathbb{R}^{(1 \times k)}$. iii) Self-product $\mathbf{A} \otimes \mathbf{A} \in \mathbb{R}^{n_A(n_A+1)/2 \times k}$ includes only unique row pairs: $\mathbf{A} \otimes \mathbf{A} = \left( \mathbf{A}_1 \otimes \mathbf{A}_1, \ldots, \mathbf{A}_1 \otimes \mathbf{A}_{n_A}, \mathbf{A}_2 \otimes \mathbf{A}_2, \ldots, \mathbf{A}_2 \otimes \mathbf{A}_{n_A}, \ldots, \mathbf{A}_{n_A} \otimes \mathbf{A}_{n_A} \right)$. iv) Note $\{0,1\} \subset \{0,1,2\} \subset \mathbb{R}$.

The term $\mathbf{s} \otimes \mathbf{x}$ captures genotype-by-cell state interactions, with $\beta$ quantifying the cell state-specific eQTL effect. We incorporated all other quadratic interaction terms as covariates, which offer several advantages for statistical inference. i) While confounders should be included to reduce false positives, the benefits of including non-confounder covariates, such as "prognostic covariates" that explain variance in the response variable, are less known. Prognostic covariates reduce residual variance and improve statistical power, even when not confounding $\mathbf{s} \otimes \mathbf{x}$[57]. One example prognostic covariate is library size, which is known to confound scRNA-seq measurements of gene expression at experiment- and cell state-dependent levels. ii) Our comprehensive inclusion of quadratic terms follows established recommendations for covariate pre-selection (even if over-selection) before any statistical inference is drawn[58]. iii) On contrary, covariate selection based on association with target variable using the same dataset risks the statistical complexities of post-selection inference[59].

Other practical aspects include replacing $\mathbf{x} \otimes \mathbf{x}$ with the presence of a dominant allele, $\mathbf{x}^{(dom)} = (x_1^{(dom)}, \ldots)$, where $x_i^{(dom)} = 0$ or 1, depending on whether $x_i \neq 0$. Since humans are biallelic ($x_i = 0, 1, 2$), $\mathbf{x} \otimes \mathbf{x}$ and $\mathbf{x}^{(dom)}$ span the same linear space together with $\mathbf{x}$ and the intercept.

The residual variance $\varepsilon \in \mathbb{R}^{n_{cell}}$ was modeled as:

$$\varepsilon \sim MVN(0, \mathbf{\Sigma}), \quad \mathbf{\Sigma} = \sigma^2(\mathbf{I} + \lambda \mathbf{K}), \tag{2}$$

where $\mathbf{K} \in \mathbb{R}^{n_{cell} \times n_{cell}}$ is the GRM, same as in Fig. 1c.

Hypothesis testing for the null hypothesis $\beta = 0$ was conducted using reML and the LRT, as in sceQTL mapping.

**SceQTL mapping—AIR accelerations.** In sceQTL mapping, AIR is used to store genotype data for individual cells in population-scale scRNA-seq studies. With its unique exact accelerations for basic matrix operations (Supplementary Note), AIR further accelerates LMMs on top of state-of-the-art implementations for each sceQTL mapping step:

- **Expanded genotype matrix.** Conventional approaches expand the donor-level $\mathbf{G} \in \mathbb{R}^{n_{SNP} \times n_{donor}}$ to cell-level $\mathbf{G}^{(E)} \in \mathbb{R}^{n_{SNP} \times n_{cell}}$ explicitly at time complexity $\mathcal{O}(n_{SNP}n_{cell})$. AIR instead stores the compressed $\mathbf{G}$ with cell counts per donor, eliminating expansion costs at $\mathcal{O}(n_{donor})$.
- **GRM estimation.** Conventional methods estimate the cell-level GRM via matrix multiplication on the expanded genotype (after mean and variance normalization) as $\mathbf{K} = (\mathbf{G}^{(E)})^T \mathbf{G}^{(E)} \in \mathbb{R}^{n_{cell} \times n_{cell}}$. This incurs a time complexity of $\mathcal{O}(n_{SNP}n_{cell}^2)$. AIR leverages efficient matrix multiplication (Supplementary Note) and reduces the time complexity to $\mathcal{O}(n_{SNP}n_{donor}^2)$.
- **GRM factorization.** A major optimization introduced in FaST-LMM and GEMMA is the eigendecomposition (via SVD) of GRM and its subsequent transformation of other matrices such as genotype, gene expression, and covariates. Conventional methods decompose the GRM $\mathbf{K} \in \mathbb{R}^{n_{cell} \times n_{cell}}$ in $\mathcal{O}(n_{cell}^2 n_{donor})$ time, even after recognizing the GRM has a rank of $n_{donor}$ due to row and column repeats. Because all cells from the same donor share the same germline genotypes, their corresponding rows and columns in $\mathbf{K}$ would be identical. This allowed AIR to merge these duplicate rows and columns with rescaled variance, further reducing eigendecomposition or SVD time complexity to $\mathcal{O}(n_{donor}^3)$ (Supplementary Note).
- **GRM transformation.** Transforming the genotype and expression matrices with GRM eigenvectors, as required in the above optimization, involves matrix multiplication. Conventional methods compute this with a time complexity of $\mathcal{O}((n_{SNP} + n_{gene})n_{cell}n_{donor})$ after accounting for the GRM's low rank. Using AIR's efficient matrix multiplication (Supplementary Note), this complexity further is reduced to $\mathcal{O}((n_{SNP} + n_{gene})n_{donor}^2 + n_{gene}n_{cell})$.
- **SNP-gene covariance.** The naive covariance calculation between genotype and gene expression (without considering GRM) takes $\mathcal{O}(n_{SNP}n_{gene}n_{cell})$ time to compute by conventional methods. AIR accelerates this computation to $\mathcal{O}(n_{gene}(n_{SNP}n_{donor} + n_{cell}))$ by leveraging its efficient matrix multiplication (Supplementary Note).

**SceQTL mapping—normalization with Normalisr.** Normalization of scRNA-seq read counts and covariates was performed following the GSE120861 tutorial for Normalisr (1.0.0). Briefly, the pipeline included the functions lcpm, normcov, scaling_factor, compute_var, and normvar from normalisr.normalisr. These steps transformed scRNA-seq read counts into the minimum mean square error (MMSE) estimator of log gene expression levels. Technical variations were extracted from the scRNA-seq read counts as additional covariates (in addition to the constant 1 intercept) for inclusion in the linear mixed model during sceQTL mapping and for variance normalization.

**SceQTL mapping—$P$ value calibration.** We calibrated $P$ values for cell state-specific sceQTL mapping using null $P$ values derived from a separately simulated null dataset (Supplementary Fig. 5). The null $P$ value distribution was modeled as $Beta(e^{\alpha_0 + \alpha_1 x}, e^{\beta_0 + \beta_1 x})$. Unlike existing approaches[37,38], we introduced $x$, one of the candidate parameters detailed below, computed separately for each gene or SNP to account for potential $P$ value distribution biases. Before calibration, $x$ was normalized to have zero mean and unit variance, and this

normalization was stored for subsequent steps. Parameters $\alpha_0$, $\alpha_1$, $\beta_0$, $\beta_1$ were estimated via maximum likelihood using stochastic gradient descent in PyTorch, with a learning rate of $10^{-6}$ over 20,000 epochs. To prevent divergence during estimation, $P$ values smaller than $10^{-300}$ were clipped to $10^{-300}$. For calibration, $x$ was computed for the original dataset, normalized as stored, and the $P$ value was adjusted by computing the cumulative distribution function (CDF) of the Beta distribution using the parameters learned above.

Candidate parameters for $x$:

- **Single-cell differential expression technical covariates**: lcpm (pseudo-bulk log(CPM+1) for each gene), n0 (proportion of cells with zero reads for each gene);
- **Derived parameters** (applied to each of the above covariates as $x$): $x$min (minimum of $x$'s values computed separately for each cell state), $x$diff (difference in $x$'s values between two cell states), $x$diffabs (absolute difference in $x$'s values between two cell states);
- **Population genetics parameters**: maf (minor allele frequency for each SNP), l0 ($\hat{\lambda}$ for each gene).

**CGRN inference—sceQTL mapping and filtering.** For cGRN inference based on sceQTL summary statistics, we first performed full cis-(within 1Mbp window before or after the transcription start site) and trans-sceQTL mapping, limiting trans-sceQTL candidate outputs to those with $P < 10^{-4}$ for computational efficiency. Gene names containing '-', '.', or '_' were excluded. SNPs not identified as cis-sceQTLs for any gene (BH $Q < 0.1$, computed separately for cis-sceQTL candidate pairs) were removed. SNPs considered potentially pleiotropic were excluded from cGRN inference unless associated with a single cis-gene (BH $Q < 0.1$) while associations with all other cis-genes were highly insignificant (raw $P > 0.1$). Among the remaining SNP-gene pairs, BH $Q$ values were calculated separately for trans-sceQTL mapping. To accomplish this, we implemented a custom function for calculating BH $Q$ values for $P$ values below a cutoff. Significant cis-sceQTLs (BH $Q < 0.1$) and trans-sceQTLs (BH $Q < 0.5$, indicating at least half of identified trans-sceQTLs were true positives) were retained as SNP-gene pairs for cGRN inference.

**CGRN inference—effect size estimation.** To estimate the effect size of each candidate causal gene regulation in each cell state, we looked for instrumental SNP → cis-gene (regulator) → trans-gene (target) relationships. Here the SNP serves as the instrumental variable, as described and identified as a cis-sceQTL for the regulator and a trans-sceQTL for the target above. Using each instrumental SNP, we estimated the gene regulation effect size as the ratio between trans- and cis-sceQTL effect sizes following 2SLS.

**CGRN inference—inference robustness evaluation.** To evaluate inference robustness, we tested how much the sign and value of estimated gene regulation effect sizes vary across instrumental SNPs. We defined the relative variability of gene regulation strength estimates as their difference between maximum and minimum values divided by the twice of maximum of absolute value. This metric gives 0 when all instrumental SNPs yield identical estimates, 0.5 when one SNP estimates no effect while the others preserve directionality, and 1 when the maximum and minimum estimates have equal absolute values but opposite signs.

**CGRN inference—filters with instrumental SNPs.** Only gene regulations with at least 2 unique instrumental SNPs were included in the cGRN. Here uniqueness is defined as having non-identical genotype data vectors between the two instrumental SNPs among the donors used to map cell state-specific sceQTLs. Differences merely due to opposite definitions of reference/alternative alleles were regarded as identical.

## Data

**Published population-scale scRNA-seq datasets.** The Randolph et al. dataset[19] contains scRNA-seq data of 255,731 peripheral blood mononuclear cells (PBMCs) sampled from 89 donors of African and European ancestry. The PBMCs were subjected to two distinct conditions: mock treatment (NI) and infection with the H1N1 Cal/04/09 influenza A virus (flu). Donors were genotyped with whole genome sequencing. Cell type and donor annotations provided in the original dataset were retained for this study. The Seurat object was provided in the original study and directly exported into the input files for airqtl without any processing. The code for reformatting is available on Zenodo under accession code 15925067.

**Quality control.** We primarily followed the quality control established for high-MOI single-cell CRISPR screen in the Normalisr study[5], with additional criteria for donor-based statistics. Briefly, we removed cells with less than 100 expressed genes or 500 reads, genes expressed in less than 10% of donors, 100 cells, or 2% cells, (cells from) donors with 5 cells or fewer, and SNPs that could not be confidently identified for any donor in the original study.

The above removal was performed separately for each cell subset (e.g., by cell type according to existing annotations) in published (and possibly already QCed) population-scale scRNA-seq data and iteratively until no gene, cell, SNP, or donor was removed. Then, cell subsets with less than 500 cells, 40 donors, 10,000 SNPs, or 500 genes remaining were excluded from sceQTL mapping and downstream analyses. Here specifically, a gene was defined as "expressed" if it exhibited at least 1 read in a particular cell or any cell from a particular donor.

**Simulation method for sceQTL mapping data.** Our simulation strategy employed permutation and resampling to maximize the similarity with real datasets for benchmarking purposes. To generate the a simulated dataset with $n_{SNP}$ SNPs, $n_{gene}$ genes, $n_{donor}$ donors, and $n_{cell}$ cells that mimics a real dataset, we followed the approach established in Normalisr[5] and introduced the contributions of cis-eQTLs. First, to resample these dimensions, we removed non-autosomal genes and randomly downsampled to $n_{gene}$ genes from the remaining ones. SNPs outside 1Mbp window of the transcriptional start site (TSS) of any remaining gene were removed and then downsampled to $n_{SNP}$ randomly to maximize cis-eQTL simulation efficiency. Then, donors and cells were randomly downsampled or upsampled to $n_{donor}$ and $n_{cell}$, respectively.

For cis-eQTL simulation, each SNP within a 1Mbp distance of any gene's TSS was randomly selected as a cis-eQTL for that gene with a 50% probability. Cis-eQTL effect sizes were then simulated by performing multivariate differential expression using Normalisr on the genotypes of all simulated cis-eQTLs in the resampled dataset. For the log expression of gene $i$ normalized via Normalisr as $y_i$ (see "SceQTL mapping -" normalization with Normalisr"), the maximum likelihood estimator for the effect size of SNP $j$ was computed as $\beta_{ji}$ within the linear model:

$$y_i = \sum_k \alpha_{ki} c_k + \sum_j \beta_{ji} x_j + \varepsilon_i^{(0)}, \tag{3}$$

where $y_i$ represents the normalized expression of gene $i$, $c_k$ denotes the normalized covariate $k$ (including the intercept), $\alpha_{ki}$ is the effect size of covariate $k$ on gene $i$'s expression, $x_j$ is the genotype of SNP $j$, and $\varepsilon_i^{(0)} \sim i.i.d\, N\left(0, \left(\sigma^{(0)}\right)^2\right)$ is the cell-level error term. To account for effect size underestimation caused by genotype correlations from linkage disequilibrium (LD) and to extend the range of effect sizes covered in benchmarking, $\beta_{ji}$ was scaled by a factor of 1.5. The $\alpha_{ki}$ term for the intercept was retained to preserve the mean expression level of gene $i$.

To simulate genotypes, the existing genotype values (0,1,2) in the resampled dataset for each SNP were randomly permuted across donors to remove LD. This allowed to benchmark false positives in cis-eQTL mapping while preserving the minor allele frequency. In this study, we regard LD a question to be addressed in fine-mapping, beyond the scope of the sceQTL mapping step. Additionally, since neither this study nor CellRegMap aimed to enhance the statistical accuracy of GRM estimation, the use of permuted genotypes provided an ideal test scenario for assessing their core functionalities without introducing unequal genetic relatedness among donors.

To simulate the (unnormalized) true log expression levels for gene $i$ in each cell, $y_i$, we extended the log expression model from Normalisr:

$$y_i = \alpha_i + \sum_j \beta_{ji} x_j + \varepsilon_i^{(0)} + \varepsilon_i^{(1)}, \tag{4}$$

where $\alpha_i$ represents the intercept and mean expression level for gene $i$, $x_j$ denotes the genotype of SNP $j$, $\beta_{ji}$ is the effect size of SNP $j$ on gene $i$, $\varepsilon_i^{(0)} \sim i.i.d \, N(0, (\sigma^{(0)})^2)$ is the cell-level error term, and $\varepsilon_i^{(1)} \sim MVN(0, (\sigma^{(1)})^2 \mathbf{K})$ is the donor-level error term. Here, $\mathbf{K}$ is a block diagonal matrix of all-one square matrices, each sized according to the number of cells from a given donor. The intercept $\alpha_i$ was estimated as the pseudo-bulk mean expression level, and $\beta_{ji}$ was estimated during the cis-eQTL simulation step. We set $\sigma^{(0)} = 0.5$ and $\sigma^{(1)} = 1$ to ensure a substantial contribution from each source of variation.

To simulate scRNA-seq read counts, we first calculated the normalized true expression level as the softmax across all genes for each cell. This step provided gene-level normalization and incorporated the exponential transformation from the unnormalized log expression level $y_i$. Then, scRNA-seq read counts for each cell were sampled randomly across all genes using a binomial distribution, with probabilities proportional to the normalized true expression levels. The total number of reads per cell was randomly sampled from its distribution derived from the resampled dataset.

Details about our simulation method can be found in airqtl.sim at (https://github.com/grnlab/airqtl).

**Simulation method for cell type-specific sceQTL mapping data.** The simulation of cell type-specific sceQTLs was based on the sceQTL simulation method with the following modifications: i) Cell type (CD4 vs. CD8 T) was included as an additional one-hot encoded covariate throughout the simulation process. Its effect size on gene expression levels was estimated in $\alpha_{ki}$ in Eq. (3) and retained for subsequent simulation. Technical covariates introduced by Normalisr were additionally normalized in advance to be orthogonal to cell type covariates, ensuring accurate effect size estimation. ii) Each cis-eQTL was assigned a 50% probability of having cell type-specific effects. For genes with cell type-specific cis-eQTLs, their expression level model (Eq. (3)) incorporated additional linear terms derived from the interaction between cell type and genotype. The corresponding effect sizes were estimated and stored for use in the simulation. iii) The simulation of true log expression levels (Eq. (4)) was updated to account for cell type-specific contributions, consistent with the modified effect size estimation model. Specifically, the two extra terms above were added along with contributions from technical covariates from Normalisr and the intercept. Their effect sizes were introduced as they were estimated above. Since these terms already characterized the mean expression levels, the $\alpha_i$ term was removed.

**Simulation method for calibration.** The simulation method for null calibration datasets of cell type-specific sceQTL mapping followed the same steps as cell type-specific sceQTL mapping benchmarking datasets but with the modifications: i) Simulations were based on

the dataset for which cell type-specific sceQTL mapping $P$ values were to be calibrated, with identical dimensions ($n_{SNP}$, $n_{gene}$, $n_{donor}$, and $n_{cell}$). Because simulated datasets were used for benchmarking, they were the input datasets for this simulation to mimic. ii) Genotypes were not permuted, as LD removal was only required for benchmarking. iii) Cis-eQTLs were present but none of them was cell type-specific. iv) The $\beta_{ji}$ values were scaled by a factor of 2 (instead of 1.5) to ensure adequate coverage of effect size ranges for calibration purposes.

**Simulated datasets for benchmarking.** To benchmark the efficiency of sceQTL mapping and cell type-specific sceQTL mapping, we generated a series of simulated datasets designed to mimic the CD4+ T cells (the largest cell subset for sceQTL mapping) or the CD4+ and CD8+ T cells (the largest and most biologically related cell subsets, for cell type-specific sceQTL mapping) from the quality-controlled real dataset by Randolph et al.[19]. Small-scale datasets, designed to accommodate the limited scalability of CellRegMap, had default dimensions of $n_{SNP} = 500$, $n_{gene} = 500$, $n_{donor} = 100$, and $n_{cell} = 1,000$ for sceQTL mapping, and $n_{SNP} = 200$, $n_{gene} = 1,000$, $n_{donor} = 50$, and $n_{cell} = 2,000$ for cell type-specific sceQTL mapping. Large-scale datasets had default dimensions of $n_{SNP} = 4,000,000$, $n_{gene} = 5,000$, $n_{donor} = 100$, and $n_{cell} = 50,000$ for sceQTL mapping, and $n_{SNP} = 200,000$, $n_{gene} = 2,000$, $n_{donor} = 100$, and $n_{cell} = 50,000$ for cell type-specific sceQTL mapping. Using these default configurations, we also generated related datasets by varying each of these dimensions to assess the scalability of each method under different scenarios.

For statistical performance benchmarking in sceQTL mapping or cell type-specific sceQTL mapping, we similarly generated one simulated dataset each with $n_{SNP} = 20,000$, $n_{donor} = 100$, $n_{cell} = 50,000$, and all post-QC genes.

In all cases, datasets were generated by regarding each donor-condition pair from the real dataset as a separate donor. This allows to expand the range of variability in the number of donors. Genotype permutations for LD removal also eliminate any residual population structure, ensuring that each simulated donor is distinct.

**Benchmarking**

**Other methods.** For CellRegMap, we followed its website and tutorials for single-cell preprocessing and association testing starting from raw read counts (https://github.com/annacuomo/CellRegMap_analyses/tree/4f26437d20e824e704385deeb2bec1d9f1f61f4e/endodiff/preprocessing, https://github.com/annacuomo/CellRegMap_analyses/blob/4f26437d20e824e704385deeb2bec1d9f1f61f4e/simulations/utils/run_tests.py, https://limix.github.io/CellRegMap/usage.html). To allow a consistent benchmarking, latent dimension covariates were not included (see below). For statistical performance benchmarking (below), only SNP-gene relations in the cis region (1Mbp up or downstream) were computed for CellRegMap's efficiency.

We did not include GASPACHO[34] and other Bayesian methods that are less compatible with downstream analyses such as fine mapping, transcriptome-wide association studies, and Mendelian randomization. We did not include BOLT-LMM[32] or other heuristic methods that rely on the same approximation such as SAIGE-QTL[24] for time complexity comparison or benchmarking. Their approximation is designed for GWAS and are prone to large errors without a large number of individuals (BOLT-LMM recommends 5000 samples), which is rarely met by current population-scale scRNA-seq studies. In addition, we could not run GASPACHO on custom datasets due to the lack of instruction on preparing input parameters. Our test run of SAIGE-QTL's tutorial also met with critical errors. Nevertheless, airqtl is estimated to run approximately $10^3$ times faster than SAIGE-QTL based on its projected running time in ref. 24. Our AIR data structure can also provide similar accelerations to these methods.

**Estimation of realistic acceleration from AIR's algorithm.** For estimation purposes, a realistic sceQTL mapping task was assumed to have 100,000 cells, 1,000 donors, 8,000 genes, and 4,000,000 SNPs post-QC for each most abundant cell type because they take up the majority of computation time. These numbers were put into the computational complexities, namely $(n_{SNP} + n_{donor})(n_{gene} + n_{cell})n_{cell}$ for FaST-LMM and GEMMA and $n_{gene}n_{cell} + (n_{SNP} + n_{donor})(n_{gene} + n_{donor})n_{donor}$ for airqtl. Their ratio yielded 1,200 as the estimated acceleration from AIR's algorithm.

**Latent dimensions as covariates.** In our benchmarking, we did not account for latent dimensions (such as those derived from PCA) from genome and/or transcriptome data as covariates for several reasons. i) This study is focused on improving computational efficiency without sacrificing statistical performance. Identifying the optimal strategy for incorporating latent factors would be the subject of a separate study. ii) While latent dimension covariates may improve false discovery rate control in sceQTL mapping, they can also obscure true signals, reduce statistical power, and lead to spurious associations[41]. Although the impact of these drawbacks varies across dimension reduction methods and data types, we would ideally prefer to identify a method that improves false discovery rate control without introducing these trade-offs. iii) Our benchmark for sceQTL mapping employed independently permuted genotypes, which lacked population structure, and yielded satisfactory results without the need for latent dimension covariates. iv) Airqtl is based on a standard linear mixed model and offers the flexibility to incorporate any type of covariate, including latent dimensions, if necessary, as demonstrated using the real dataset from Randolph et al.

**Efficiency benchmarking.** We compared the efficiency of CellRegMap and airqtl on Google Cloud Compute Engine, using the machine type for which each method is designed, and recorded the running times with Snakemake. Specifically, airqtl was run on a2-highgpu-1g with 1 NVIDIA A100 40GB GPU. Since each CellRegMap process was single-threaded, we ran up to 32 CellRegMap processes in parallel on n2d-highmem-32 with 32 vCPUs. No resource competition for vCPUs or memory was observed among the 32 parallel processes. The computing cost was estimated by multiplying the running time by the spot price of the Google Cloud Compute Engine instance ($1.469/hour for a2-highgpu-1g and $0.668/hour for n2d-highmem-32). The cost for CellRegMap was further divided by 32, as each process utilized only 1 vCPU, and the price was linearly proportional to the number of vCPUs. The datasets described in "Simulated datasets for benchmarking" were used for these evaluations.

Since CellRegMap could not scale to the large datasets, we projected its running time and associated cost based on its performance on smaller datasets, using the following linear model:

$$\log t = \alpha + \beta_{n_{SNP}} \log n_{SNP} + \beta_{n_{gene}} \log n_{gene} + \beta_{n_{donor}} \log n_{donor} + \beta_{n_{cell}} \log n_{cell} + \varepsilon, \tag{5}$$

where $\varepsilon \sim N(0, \sigma^2)$. For sceQTL mapping, all data points for CellRegMap in 2a were used to train this model and project the running times for CellRegMap in 2b. For cell type-specific sceQTL mapping, all data points for CellRegMap in Fig. 3a were used to train the model and project the running times for CellRegMap in Fig. 3b.

**Statistical performance benchmarking.** We used the datasets described in "Simulated datasets for benchmarking" for statistical performance benchmarking. We benchmarked the estimation accuracy of sceQTL effect sizes and cell type-specific sceQTL effect sizes with bias and variance, following[5]. Briefly, bias characterizes the overall over- or under-estimation of effect sizes and can be quantified in the

linear model:

$$y = \beta x + \varepsilon, \ \varepsilon \sim N(0, \sigma^2), \tag{6}$$

where $y$ represents the estimated effect sizes, and $x$ denotes the ground-truth effect sizes for cis-eQTLs in the simulated dataset. Therefore, $\beta$ reflects the estimated effect size scale. The maximum likelihood estimator for $\beta$ was computed as $\hat{\beta}$, and its deviation from 1 indicates systematic bias. Variance was estimated as the deviation of Pearson $R$ from 1 between the estimated and ground-truth effect sizes. Given that expression-dependent underestimation of effect sizes is commonly observed in single-cell differential expression[5], a simpler version of sceQTL mapping, we performed this benchmarking both for all genes together and for genes stratified by expression level (pseudo-bulk logCPM) to assess the severity of this problem in sceQTL mapping and cell type-specific sceQTL mapping.

To benchmark the specificity of each method, we separately compared the sceQTL $P$ value distribution for non-eQTLs or non-specific sceQTLs against the uniform distribution. To assess the sensitivity of each sceQTL mapping method, we compared the $P$ value distribution of eQTLs against those of non-eQTLs. Similarly, to evaluate the sensitivity of each cell type-specific sceQTL mapping method, we compared the $P$ value distribution of cell type-specific eQTLs against those of non-eQTLs and non-specific eQTLs.

**Type I error evaluations.** We drew $P$ value histograms for sceQTL mapping or cell type-specific sceQTL mapping $P$ values for SNP-gene pairs categorized as non-eQTLs (SNP has no effect on gene expression), eQTLs (SNP has a nonzero effect on gene expression), non-specific eQTLs (SNP has a nonzero effect on gene expression independent of cell type), or cell type-specific eQTLs (SNP has a nonzero effect on gene expression dependent on cell type). The ground-truth eQTL status of these pairs was determined in the simulation data. Each histogram was divided into 20 equal bins from 0 to 1. Error bars were estimated as $3\sqrt{N}$, where $N$ is the number of entries in each bin. Dashed lines represent the uniform distribution.

We used two-sided Kolmogorov-Smirnov test against the standard uniform distribution to evaluate the specificity of each method. We also used Storey's $\pi_1$ estimated by fdrtool[60] (1.2.18) to characterize the proportion of false positives identified by each method or configuration in fully null datasets.

**State-specific cGRN inference and analysis**
**Population admixture detection and removal.** We performed a Pearson correlation test between SNP pairs located on different chromosomes to avoid correlation due to linkage disequilibrium. To accelerate computation, only 100,000 post-QC SNPs were randomly sampled for this test. As a baseline, we also performed this test after permuting the donors separately for each SNP.

To test the effect of population admixture removal on genotype correlation, we introduced different numbers of top PCs of the genotype matrix for the tested SNPs as covariates for the (conditional) Pearson test.

To test the effect of population admixture removal on trans-sceQTL mapping, we used GRM estimated from the genotype matrix and/or introduced its the top principal component from the genotype matrix as a covariate without LD pruning.

**Comparison of cell state-specific cGRNs.** The overlap rate was first computed as the ratio between the number of overlapping gene regulations and the minimum number of gene regulations between the two cGRNs being compared. Overlapping gene regulations between two cell states were defined as those present in both cGRNs, irrespective of effect size or directionality estimation. Any deviation of the overlap rate from one may reflect contributions from differences in the

cGRNs, differences in the detectability of individual gene regulations, and data randomness due to the finite number of cells.

Among the overlapping gene regulations, three metrics were computed to characterize the actual differences in cGRNs. The directionality difference was defined as the proportion of gene regulations that exhibited opposite signs in the estimated effect sizes. Pearson $R$ was calculated for the estimated effect sizes between the two cell states to assess the overall similarity. Regulation scale was computed as the linear regression coefficient between the estimated effect sizes of the two cell states (inverted if $< 1$). Since only overlapping gene regulations were considered, these metrics are less influenced by factors other than differences in the cGRNs.

**Visualization of cell state-specific cGRNs.** We used Cytoscape (3.10.2) for cGRN visualization. Prefused force directed layout was applied on the largest connected component of the cGRN for CD4+ T cells under flu condition using regulation strength as edge weight.

**Gene ontology enrichment analysis.** For each master regulator, we used GOATOOLS (1.3.11) for GO enrichment analysis on its target genes using all other genes that have a regulator in the cGRN as background. This reduces the bias of background gene selection compared to using all genes or genes expressed in the corresponding cell state because they do not account for cell state specificity of cGRNs. Activated targets (with regulation strength > 0) and repressed targets (with regulation strength < 0) were analyzed for GO enrichment separately. To avoid evaluation biases for our expression-based analyses, for all GO analyses we excluded GO evidences that may also have an expression-based origin (IEP, HEP, RCA, TAS, NAS, IC, ND, and IEA). All three GO categories were used. Both over- and under-enrichments were calculated. Gene names were converted with mygene (3.2.2) when needed. Basic ontology (2024-09-08) and human GO association (2024-09-10) files were used.

**Query of perturbation experiments.** We queried GEO for perturbation experiments involving any of the 10 master regulators in CD4+ T cells under the flu condition. The experiments were required to involve primary CD4+ T cells or their derived cell lines. Perturbations should include overexpression, knockdown, or knockout of the master regulator, along with a control condition, followed by either bulk or single-cell RNA-seq or microarray measurements. Viral infection or exposure was not a requirement. Only target genes that were significantly identified in the perturbation experiment and also found in the inferred cGRN were included for comparison. Perturbation experiments with ≤ 5 overlapped targets were excluded from the comparison.

**Analyses of ChIP-seq experiments.** We queried the whole GEO for ChIP-seq experiments of STAT1 in CD4+ T cells in human and found one dataset with two conditions: INFg treated and control.

For each condition, we performed liftover of their annotated STAT1 peaks from hg18 (reference genome in the original study) to hg38 (reference genome in Raldoph et al.). We used Homer (version 5.1, scanMotifGenomeWide function) to filter high-confidence STAT1 binding sites by filtering only peaks containing canonical STAT1 motifs. Each gene was then annotated with its nearest STAT1 binding site based on transcription start site (TSS) proximity.

For the Venn diagrams, we used a wide range of distance thresholds from 0.02 to 500kb as a parameter to determine whether each gene was regarded a STAT1 direct target. A hypergeometric test was performed to assess the statistical significance of the overlap, using all genes with at least one regulator gene in the cGRN as the background. For the boxplots, we used 5kb and 100kb for thresholds representing proximal and distal regulation activity. To account for potential target gene sharing among flu-condition master regulators, we performed our comparisons using NI-condition master regulators as the baseline reference when assessing overlap with ChIP-seq identified targets.

## Conventions
All $P$ values are two-sided raw unless otherwise specified. For boxplots, boxes extend between 1st and 3rd quartiles. Lines indicate median. Whiskers extend from the box to the farthest data point within 1.5x of the inter-quartile range from the box, unless otherwise stated. Fliers are shown if beyond the whiskers.

## Reporting summary
Further information on research design is available in the Nature Portfolio Reporting Summary linked to this article.

## Data availability
The Randolph et al. population-scale scRNA-seq dataset is publicly available on Zenodo under accession code 4273999 (https://doi.org/10.5281/zenodo.4273999[61]). The transcription factor and motif list was downloaded from (https://resources.aertslab.org/cistarget/motif_collections/v10nr_clust_public/snapshots/motifs-v10-nr.hgnc-m0.00001-o0.0.tbl) The differential expression analysis result of *STAT1* perturbation was downloaded from Supplementary Data of[54]. The ChIP-seq of STAT1 was downloaded from GEO under accession GSE27158.

## Code availability
Airqtl (0.2.0) is publicly available and has been deposited in GitHub at (https://github.com/grnlab/airqtl), under BSD-3 license. The specific version of the code associated with this publication is archived in Zenodo under accession code 16538496 (https://doi.org/10.5281/zenodo.16538496) and is accessible via[62]. Users are permitted to reuse, modify, and distribute the code in accordance with the terms of the license. Any modifications to the code should appropriately credit the original authors as outlined by the license terms. The code to format Randolph et al. dataset to airqtl input is available on Zenodo under accession code 15925067 (https://doi.org/10.5281/zenodo.15925067[63]).

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

## Acknowledgements

LW wishes to thank Heping Xu and Kirk Gosik for their early support and Xinghu Qin for helpful discussions. This work was supported by National Institutes of Health grant R35GM160536 (LW).

## Author contributions

M.F.: software, methodology, validation, formal analysis, investigation, visualization, writing. Y.W.: methodology, software, validation, data curation, writing. L.W.: conceptualization, methodology, software, validation, formal analysis, investigation, data curation, writing, visualization, supervision.

## Competing interests

The authors declare no competing interests.
