## [Transparent Peer Review file · Nature Communications]

Airqtl dissects cell state-specific causal gene regulatory networks with efficient single-cell eQTL mapping

Corresponding Author: Dr Lingfei Wang

Version 0:

Reviewer comments:

Reviewer #1

(Remarks to the Author)

Lingfei Wang presents a very efficient implementation for the detection of single-cell gene expression QTLs (sceQTL), which is called Airqtl. At its heart, Airqtl performs greatly accelerated matrix operations utilizing GPU functionalities. AIR stands for Array of Interleaved Repeats, which exploits the fact that all cells from the same donor have the same genotype. Further it uses torch.Tensor to perform the matrix operations in GPUs. Importantly, all of these computations are performed while correcting for population-substructure using linear mixed models (LMM), which is currently state-of-the-art for QTL mapping in human populations.

The submission presents a significant methodological advancement and it is mostly very clearly written. I only have a few, mostly minor comments.

1. The core of the method should be better described. The paragraph on AIR in the Methods section is too brief to really understand what it is doing. The text in the Methods only says what AIR does, but provides hardly any details on how it does it. One has to refer to the supplement to understand the approach. I suggest to provide a more concrete description of how AIR works and in which way it exploits functionalities of torch.Tensors in the Methods section.
2. Related to the first point: it should be better explained in which way AIR is different from pseudo-bulking. This has to become more clear already in the Results section ("Airqtl overview"). Note that also in Figure 1a, green box "AIR acceleration" it is not really clear why this is not simply pseudo-bulking per donor. The matrix on the right hand side (Gene x Donor) looks as if all cells of a donor were collapsed into a single vector. This point also has to be improved.
3. Line 20: T is missing at the beginning of the sentence
4. Figure 2 & 3, specify in the description/text (not only in methods) the values at which the constants n were kept.
5. Figure 3e: First of all, the shades (representing confidence intervals) are hardly visible. Second, the confidence intervals seem very small given the variation (noise?) of the estimates along the x-axis. I can't believe that all of the small 'wiggles' would be 'true', which is suggested by the super small confidence intervals.
6. Figure 4c: The differences between the models with and without PC1 are hardly visible; colors/dots are overlapping too much. Consider a scatter plot of p-values with versus without the 1st PC.
7. Page 11: 'Regulation scale' needs to be explained here. Unlike Pearson's R, this is not a commonly known measure.
8. Page 13, lines 214 - 219: Does the author interpret this as a cell-type dependent 're-wiring' of the network (alternative network topology) or is this rather a cell-type-dependent activation/repression of target genes? In other words: to what extent is the topology of the network actually changing as opposed to altering the activity of different parts of an otherwise static network topology?
9. It would be very helpful to describe example 1.3 in the Supplementary Note in more detail and apply it to the definition above. Then it would also be easier to understand how the repeats are reduced to one representation.

10. Suppl. Fig. 7: I am not sure this figure conveys any relevant information. In the main text it reads "... with upregulated genes clearly enriched in one cluster". I don't see this from this figure. Maybe a heatmap showing the clustered adjacency matrix might work better?

(Remarks on code availability)

Reviewer #2

(Remarks to the Author)

This manuscript describes a new computational approach, AIRQTL, for single-cell-based eQTL mapping. By changing the strategy for eQTL mapping—usually based on SNP vs. cell and cell vs. gene—to SNP × donor and donor × gene matrices, AIR improves the computational efficiency of sc-eQTL mapping. While the computational improvements are clear, it is not evident how the eQTL estimates produced by the new algorithm advance the eQTL analysis problem. Additionally, the comparative analysis with alternative competing methods is limited and should be expanded and improved.

Main Points:

I could not find results supporting the statement: "Compared to state-of-the-art LMM methods such as FaST-LMM [27] and GEMMA [28], which operate at $O((n_{\text{SNP}} + n_{\text{donor}})(n_{\text{gene}} + n_{\text{cell}})n_{\text{cell}})$ time complexity, AIR delivers a ~1000-fold algorithmic speedup for sc-eQTL mapping with 100,000 cells and 1,000 donors per cell type." Currently, only a computational complexity analysis is provided in the supplement and an empirical evaluation with CellRegMap is included.

The authors should also contrast their results with other similar approaches. How does the method compare with bulk-based approaches or cell-type bulking approaches? There are at least two other tools for sc-eQTL analysis in the literature (references 33, 34), which should also be discussed and compared in the manuscript.

Why is CellRegMap not evaluated in Fig. 3c–e? The authors should clarify this omission or include the comparison.

The authors discuss a conceptual improvement (line 114) potentially arising from the use of Normalizer. They should conduct ablation experiments to support this claim. Additionally, they could demonstrate how this improvement enhances cell-type-specific eQTL detection.

Overall, it is difficult to determine whether AIRQTL offers a computational speed improvement, a conceptual improvement, or both. The manuscript should clarify how AIRQTL compares with current state-of-the-art methods in these respects.

Line 206: Do the results imply that there are 641 proteins potentially acting as transcription factors (TFs), although their functions are currently unknown? The authors should investigate the potential functions of these proteins—e.g., whether they preferentially interact with known TFs or are associated with chromatin-related functions. Similarly, more insight into the functional roles of the proteins in the gene regulatory networks (GRNs) shown in Fig. 5d is needed.

Regarding the GRN in Fig. 5, STAT1 is shown as a potential regulator. Since STAT1 is a TF, the authors are encouraged to investigate whether the predicted interactions are supported by TF binding data (e.g., ChIP-seq or TF binding motif analysis).

There is a lack of detail on how the input data for AIRQTL was generated from the Randolph et al. dataset (line 383). This section should be expanded and clarified.

We could run the software, but there is limited documentation and it mostly consists of scripts to re-run the experiments. This should be improved. Here are some additional comments.

The Airqtl code is clean and well-structured, with good comments on the parameters.

There is no reference for the python version needed to run.

Data format can only be understood via a script. Author is encouraged to make use of Data Containers (AnnData / scanpy). This would make the usability of the method possible for a broader audience.

The method uses PCA reduced dimensions of the transcriptome data as covariates in the linear mixed model to control hidden structures. The package requires a covariates matrix as input, which can be obtained from the expression matrix.

There is no option/output to visualize cGRNs.

Minor Points:

In Fig. 1, it is unclear how AIRQTL obtains cell-type-specific eQTL estimates. This should be clarified in the figure or its legend.

Line 196 mentions chromatin rewiring. What data supports this claim? Was chromatin directly analyzed? If not, the authors should rephrase this statement to more accurately reflect the observed results (e.g., “rewiring of the GRN”).

Line 259 should be rephrased. While eQTL analysis offers a robust statistical framework for linking genetic variants to expression traits, the manuscript does not provide or cite supporting evidence for this claim.

(Remarks on code availability)

See main comments.

Reviewer #3

(Remarks to the Author)

This paper presents Airqtl, a novel computational method for mapping single-cell expression quantitative trait loci (sceQTLs) to infer cell state-specific causal gene regulatory networks (cGRNs). The motivation stems from the computational limitation of existing methods that are unable to scale to population-level scRNA-seq studies. By employing a compressed data structure called Array of Interleaved Repeats (AIR) together with GPU and other implementation optimization, airqtl significantly improves computational efficiency, achieving orders of magnitude acceleration compared to existing methods. The scalability of airqtl facilitates cell state-specific sceQTL mapping and allows robust and experimentally validated cGRNs. The author benchmarked airqtl against existing methods using both simulated and real datasets, and the real data analyses inferred cGRNs consistent with established biological knowledge.

This work makes contributions to sceQTL mapping and cGRN inference. However, I still have some major and minor comments listed below that may be helpful to improve the study.

Major comments:

1. Since computation acceleration is a major contribution of this work, the related computational complexity parts need further clarity and accuracy. In the 'SceQTL mapping — algorithmic accelerations' section, 1) for GRM estimation, $G(E)$ is a matrix of n_{snp} by n_{cell} , so the complexity of $G(E)$ multiplying its transpose should be $O(n_{\text{snp}}^2 * n_{\text{cell}})$ instead of $O(n_{\text{snp}} * n_{\text{cell}}^2)$. The same issue occurred to GRM for G of size n_{snp} by n_{donor} . 2) In general, decomposition of matrix n by n requires a complexity of $O(n^3)$. Can the author elaborate on why GRM decomposition for the n_{snp} by n_{snp} matrix $G(E) * G(E)^T$ requires $O(n_{\text{cell}}^2 * n_{\text{donor}})$ and ARI requires $O(n_{\text{donor}}^3)$? 3) The GRM transformation and SNP-gene covariance parts also need further clarification regarding the computational complexity.
2. As computational complexity was discussed in the paper, fig 2 (a) and (b) are somewhat misleading, as they appear to suggest that the computational cost scales linearly with all dimension parameters. A brief discussion of the results on the practical and theoretical scalability should be added here.
3. In Fig. 1(a), if $E(R) = \sum_{\text{in_donor}}(E)$ denotes summation over all cells for a specific donor, is there information loss here (biologically or statistically), in comparison to using the cell-specific expression data? Can the author elaborate on this point?
4. In fig. 1, since (d) uses data from (c), the readability can be improved by connecting the two parts with shared notations. For instance, using C to denote the normalized covariates in both (c) and (d).
5. Comparing fig. 2(a) with 2(b), the second and third columns are a bit redundant as the numbers genes and donors do not change too much. Also, it would be good to clarify the fixed sized when one dimension size changes.
6. The accuracy of section 'SceQTL mapping — linear mixed model for cell state specificity' needs further improved or at least additional clarification. If A is a matrix of n_A by k , B is a matrix of n_B by k , the Kronecker product is of $n_A * n_B$ by k^2 . The author also needs to clarify the self-product case.
7. Can the author elaborate on why an LMM with all linear and quadratic interaction terms was adopted? Was such quadratic interaction reflected in the result? How to interpret both statistically and biologically?
8. For equation (1), detailed description, including the dimension of each notation and their practical meaning, is needed to improve the readability of the manuscript.

Minor comments:

1. On page 2 line 20, there is a typo in 'hese approaches ...'.
2. For better comparison, fig 2(c) and 2(d) should be combined with different colors denoting different methods.

(Remarks on code availability)

Version 1:

Reviewer comments:

Reviewer #2

(Remarks to the Author)

The authors have clarified most of my requests and clarified the advances of Airqtl lies on the computational speed up, which allows the proposed analysis for the first time. They also explained (mostly technical reasons) why a broader benchmarking was not feasible. This resolves most of my requests. I do have a pending request regarding the stat1 chip-seq analysis.

Specific point

While the overlap analysis (Sup. Fig. 12) suggests some relation between Stat1 targets and ChiP-seq targets; this analysis lacks statistical assessment. Authors could check the overlap for randomly selected genes (same size as Stat1 targets) and estimate the enrichment with a hypergeometric test (or similar).

(Remarks on code availability)

Reviewer #3

(Remarks to the Author)

The author has addressed all my comments. I do not have any further comments.

(Remarks on code availability)

We sincerely thank all reviewers for their thoughtful and constructive feedback, which has significantly improved our manuscript. Below, we provide a high-level summary of key shared concerns and our responses before addressing individual reviewer comments in detail. Our response and proposed changes to the manuscript are marked in blue.

Shared Key Concerns and Responses:

1. Clarification of AIR's novelty and distinction from pseudo-bulking

All Reviewers raised questions about how AIR differs conceptually from pseudo-bulking and how it achieves computational acceleration without information loss. We have:

- Expanded Results, Methods, and Supplementary Note to provide both high-level and technical descriptions on AIR's exact algorithmic acceleration and its application to sceQTL mapping.
- Explained four major distinctions between AIR and pseudo-bulking: exact acceleration, statistical power, statistical modelling, and implementation.
- Revised Figure 1a to highlight AIR's role in accelerating multiple LMM steps, not just SNP-gene covariance calculations.

2. Benchmarking and validation of computational improvements

Reviewers #2 and #3 requested additional clarification and validation of AIR's claimed speedup and comparisons to more methods. We have:

- Added a new Methods section and revised Results to estimate realistic acceleration provided by airqtl.
- Clarified limitations in benchmarking other tools (e.g., SAIGE-QTL, GASPACHO) due to technical challenges.
- Included ablation experiments to compare with LogCPM as baseline normalization.

Reviewer #1 (Remarks to the Author):

Lingfei Wang presents a very efficient implementation for the detection of single-cell gene expression QTLs (sceQTL), which is called Airqtl. At its heart, Airqtl performs greatly accelerated matrix operations utilizing GPU functionalities. AIR stands for Array of Interleaved Repeats, which exploits the fact that all cells from the same donor have the same genotype. Further it uses torch.Tensor to perform the matrix operations in GPUs. Importantly, all of these computations are performed while correcting for population-substructure using linear mixed models (LMM), which is currently state-of-the-art for QTL mapping in human populations.

The submission presents a significant methodological advancement and it is mostly very clearly written. I only have a few, mostly minor comments.

We appreciate Reviewer #1 for their positive comments.

1. The core of the method should be better described. The paragraph on AIR in the Methods section is too brief to really understand what it is doing. The text in the Methods only says what AIR does, but provides hardly any details on how it does it. One has to refer to the supplement to understand the approach. I suggest to provide a more concrete description of how AIR works and in which way it exploits functionalities of torch.Tensors in the Methods section.

We thank Reviewer #1 for this suggestion. We have expanded the Methods section to provide a more concrete description of how AIR works and in which way it exploits functionalities of torch.Tensors (Lines 283-292):

AIR is a compressed data structure optimized for matrices containing repeated rows or columns, a common pattern in hierarchical data structures. Rather than fully expanding matrices, AIR maintains: i) the compressed matrix containing only one copy of each repeated rows/columns and ii) the number of repeats. This design enables faster matrix operations (addition, multiplication, singular value decomposition or SVD) by operating directly on compressed matrices. The acceleration leverages several mathematical properties including computing compressed results, variance rescaling, and other operand preprocessing. Based on and compatible with torch.Tensor, AIR executes all matrix operations within pytorch, simultaneously benefiting from AIR's novel algorithms and pytorch's efficient GPU acceleration.

See Supplementary Note for algorithmic details. See "SceQTL mapping --- AIR accelerations" in Methods below for AIR's application to sceQTL mapping acceleration.

2. Related to the first point: it should be better explained in which way AIR is different from pseudo-bulking. This has to become more clear already in the Results section ("Airqtl overview"). Note that also in Figure 1a, green box "AIR acceleration" it is not really clear why this is not simply pseudo-bulking per donor. The matrix on the right hand side (Gene x Donor) looks as if all cells of a donor were collapsed into a single vector. This point also has to be improved.

We very appreciate Reviewer #1 for pointing out this potential confusion. AIR differs fundamentally from pseudo-bulking in four key aspects:

i) Exact acceleration: AIR provides exact algorithmic acceleration, producing identical results to single-cell eQTL mapping that does not use AIR, while pseudo-bulking incurs information loss.

ii) Statistical power: AIR maintains the effective sample size as total cell count and preserves statistical power, unlike pseudo-bulking which reduces to donor count as demonstrated in SAIGE-QTL and other studies.

iii) Statistical modelling: AIR can account for cell level covariates and cell count imbalance between donors or contexts, which cannot be achieved by pseudo-bulking.

iv) Implementation: even in the SNP-gene covariance calculation, AIR performs log normalization on cell-level expression levels before aggregation, while pseudo-bulking aggregates expression read counts before log normalization, leading to different results.

To avoid this confusion, we have revised the Results section and focused on how AIR accelerates several steps (Lines 64-72):

The core of airqtl is Array of Interleaved Repeats (AIR), a compressed data structure designed to efficiently handle repeated values, a pattern common in sceQTL mapping (Fig. 1a). SceQTL mapping requires genotype matrix expansion from donor- to cell-level, leading to repeated values that incur substantial computational and storage overhead. AIR circumvents this inefficiency by storing only their original values and repeat patterns, enabling novel exact algorithms to accelerate basic matrix operations essential for sceQTL mapping, such as multiplication and singular value decomposition (Supplementary Note).

For instance, AIR factorizes cell-level genetic relationship matrix (GRM) in $O(n_{donor}^3)$ time complexity, compared to $O(n_{donor}n_{cell}^2)$ required by state-of-the-art methods like FaST-LMM [28] and GEMMA [29]. This principle underpins similar accelerations across all linear mixed model (LMM) steps in sceQTL mapping [...]

The revised Figure 1a now highlights how AIR algorithmically accelerates multiple computational steps:

Figure 1: Overview of algorithmic acceleration from AIR and sceQTL mapping with airqtl.

a AIR acceleration in key LMM steps. Existing methods expand the genotype matrix from donor-level (color) to cell-level, creating repeated rows and columns (dots connected by edges) that incur computational inefficiency. AIR instead operates on the compressed matrix and the number of repeats (2 3 1 shown), enabling mathematically equivalent but more efficient computations. Time complexity for each method (box color) is indicated below each step. [...] Box color (ab): method (yellow: existing; green: AIR/airqtl; gray: data). Matrix color (a) and schematic low-dimensional embedding color (b): donor. Each entry in the GRM K (a) can be shaded in up to two colors because its row and column indices can be associated with two different donors.

3. Line 20: T is missing at the beginning of the sentence

We appreciate Reviewer #1 for identifying this typo. We have corrected this typo in Line 20.

4. Figure 2 & 3, specify in the description/text (not only in methods) the values at which the constants n were kept.

We thank Reviewer #1 for this suggestion. We have included default values of n in the figure legends, which now read:

Fig 2ab: Running time (left Y) and computing cost (right Y) for sceQTL mapping on small-scale (a, defaults to $n_{SNP}=500$, $n_{gene}=500$, $n_{donor}=100$, $n_{cell}=1,000$) and realistic large-scale (b, defaults to $n_{SNP}=4,000,000$, $n_{gene}=5,000$, $n_{donor}=100$, $n_{cell}=50,000$) datasets with varying dimension sizes (X).

Fig 3ab: Running time (left Y) and computing cost (right Y) for cell type-specific sceQTL mapping on small-scale (a, defaults to $n_{SNP}=200$, $n_{gene}=1,000$, $n_{donor}=50$, $n_{cell}=2,000$) and large-scale (b, defaults to $n_{SNP}=200,000$, $n_{gene}=2,000$, $n_{donor}=100$, $n_{cell}=50,000$) datasets with varying dimensions. Color: method.

5. Figure 3e: First of all, the shades (representing confidence intervals) are hardly visible. Second, the confidence intervals seem very small given the variation (noise?) of the estimates along the x-axis. I can't believe that all of the small 'wiggles' would be 'true', which is suggested by the super small confidence intervals.

We thank Reviewer #1 for this important observation. We agree that the small wiggles are not true. To improve visibility, we reduced the transparency of shades and increased the error widths from $2\sqrt{N}$ to $3\sqrt{N}$ for all histograms with error bars (Fig 3e, S Figs 1ab, 4). Improved visibility from new figures clearly indicates the wiggles are within the error bars:

Figure 3: Airqtl enables efficient cell type-specific sceQTL mapping, comprehensive benchmarking, and objective optimization.

e Airqtl's scalability enabled benchmarking and optimization of P-value calibration for cell type specificity. Null P-value histograms are shown for non-eQTLs (left) and non-specific eQTLs (right) before (raw) and after (calibrated) calibration. Shades in histograms represent error bars ($\frac{1}{\sqrt{N}}$, where N is the number of entries in each bin). Dashed line: perfect performance (standard uniform distribution).

6. Figure 4c: The differences between the models with and without PC1 are hardly visible; colors/dots are overlapping too much. Consider a scatter plot of p-values with versus without the 1st PC.

We thank Reviewer #1 for noticing this. As noted in our original manuscript (old Lines 171-172), the differences between the models with and without PC1 are indeed hardly visible. While we acknowledge that a scatter plot comparison might better visualize subtle differences, this level of detail appears orthogonal to the conclusions made in this paper. It remains unclear to us what new conclusions can be drawn from this difference.

Having said that, we agree with Reviewer #1 on the added value in explicitly mentioning the high overlap with and without PC1. We have expanded our discussion of these results (Lines 173-175):

Notably, genotype-based GRMs had almost identical performances with/without PCs in the QQ plots. This indicated little impact from adding PC covariates, possibly because GRMs already effectively account for admixed population structure.

7. Page 11: 'Regulation scale' needs to be explained here. Unlike Pearson's R, this is not a commonly known measure.

We thank Reviewer #1 for identifying this terminology gap. We have included a high-level description (Lines 200-202):

We further quantified changes in regulation strength with regulation scale and Pearson R, reflecting differences in overall magnitude and individual gene regulations respectively, akin to our benchmarks for sceQTL effect size estimation (Fig. 2ef and Fig. 3cd, Methods).

The technical definition of regulation scale is included in Methods (old Lines 579-580, new Lines 631-632).

8. Page 13, lines 214 - 219: Does the author interpret this as a cell-type dependent 're-wiring' of the network (alternative network topology) or is this rather a cell-type-dependent activation/repression of target genes? In other words: to what extent is the topology of the network actually changing as opposed to altering the activity of different parts of an otherwise static network topology?

We thank Review #1 for this insightful question. Our results support changes in the topology of the network, as we have discussed in Lines 221-223:

Since all master regulators and many target genes remain expressed in both conditions (Supplementary Data 4), the cell state specificity is consistent with changes in gene regulation (cGRN edge) instead of (in)activation of individual genes (cGRN node).

We also noted in Discussion (Lines 262-263):

A promising future direction would be representing cell state-specific linear cGRNs as a unified nonlinear cGRN operating on diverse expression states.

9. It would be very helpful to describe example 1.3 in the Supplementary Note in more detail and apply it to the definition above. Then it would also be easier to understand how the repeats are reduced to one representation.

We appreciate Reviewer #1's suggestion to elaborate on the technical details of AIR acceleration. In response, we have significantly expanded example 1.4 (formerly 1.3) in the Supplementary Note. The Methods section "SceQTL mapping --- AIR accelerations" (Lines 339-366) now includes additional high-level descriptions.

10. Suppl. Fig. 7: I am not sure this figure conveys any relevant information. In the main text is reads "... with upregulated genes clearly enriched in one cluster". I don't see this from this figure. Maybe a heatmap showing the clustered adjacency matrix might work better?

We thank Reviewer #1 for this suggestion. We agree our statement is not immediately clear in Suppl. Fig. 7. However, we also believe 2D visualization is an integral part of gene regulatory network studies, whose absence cannot be fully compensated by heatmaps. To improve the clarity and guide readers, we have added the following sentences in the legend:

Upregulated genes (red nodes) are enriched in one cluster (right). Magnification is recommended to distinguish red nodes from red edges.

Reviewer #2 (Remarks to the Author):

This manuscript describes a new computational approach, AIRQTL, for single-cell-based eQTL mapping. By changing the strategy for eQTL mapping—usually based on SNP vs. cell and cell vs. gene—to SNP \times donor and donor \times gene matrices, AIR improves the computational efficiency of sc-eQTL mapping. While the computational improvements are clear, it is not evident how the eQTL estimates produced by the new algorithm advance the eQTL analysis problem. Additionally, the comparative analysis with alternative competing methods is limited and should be expanded and improved.

We appreciate Reviewer #2's positive comments confirming our computational improvements.

Main Points:

1. I could not find results supporting the statement: "Compared to state-of-the-art LMM methods such as FaST-LMM [27] and GEMMA [28], which operate at $O((n_{\text{SNP}} + n_{\text{donor}})(n_{\text{gene}} + n_{\text{cell}})n_{\text{cell}})$ time complexity, AIR delivers a ~ 1000 -fold algorithmic speedup for sc-eQTL mapping with 100,000 cells and 1,000 donors per cell type." Currently, only a computational complexity analysis is provided in the supplement and an empirical evaluation with CellRegMap is included.

We thank Reviewer #2 for requesting supporting information. In response, we added a new Methods section "Estimation of realistic acceleration from AIR's algorithm" (Lines 546-551):

For estimation purposes, a realistic sceQTL mapping task was assumed to have 100,000 cells, 1,000 donors, 8,000 genes, and 4,000,000 SNPs post-QC for each most abundant cell type because they take up the majority of computation time. These numbers were put into the computational complexities, namely $(n_{\text{SNP}}+n_{\text{donor}})(n_{\text{gene}}+n_{\text{cell}})n_{\text{cell}}$ for FaST-LMM and GEMMA and $n_{\text{gene}}n_{\text{cell}}+(n_{\text{SNP}}+n_{\text{donor}})(n_{\text{gene}}+n_{\text{donor}})n_{\text{donor}}$ for airctl. Their ratio yielded 1,200 as the estimated acceleration from AIR's algorithm.

2. The authors should also contrast their results with other similar approaches. How does the method compare with bulk-based approaches or cell-type bulking approaches? There are at least two other tools for sc-eQTL analysis in the literature (references 33, 34), which should also be discussed and compared in the manuscript.

We thank Reviewer #2 for suggesting broader comparisons.

Bulk-based approaches or cell-type bulking approaches have been demonstrated to suffer severe limitations compared to their single-cell counterparts. We have updated our Introduction and provided references to compare sceQTL mapping with pseudo-bulk strategies in Lines 47-48:

Some studies employ pseudo-bulk strategies for eQTL mapping, but they face challenges with continuous cell states, technical confounders, and limited statistical power [15, 20, 24].

Regarding benchmarking two other tools for sc-eQTL analysis in the literature (references 33, 34), we encountered critical errors while running SAIGE-QTL and could not determine some parameters for GASPACHO. The authors did not provide support to resolve these issues (<https://github.com/natsuhiko/GASPACHO/issues/3> and

<https://github.com/weizhou0/qtl/issues/24>). We reported our attempts and detailed other reasons of exclusion in Lines 101-104:

We excluded other methods such as GASPACHO [34] and SAIGE-QTL [24] for three reasons: i) their Bayesian or heuristic frameworks reduce compatibility with downstream analyses or population-scale scRNA-seq datasets (Methods); ii) they lack formal statistical tests for sceQTL context specificity; iii) we encountered critical challenges while trying to run these methods on official or custom datasets.

3. Why is CellRegMap not evaluated in Fig. 3c–e? The authors should clarify this omission or include the comparison.

We appreciate Reviewer #2's query regarding CellRegMap comparisons. We already clarified this omission in our initial submission (old Lines 133-134 and new Lines 135-136):

Although CellRegMap's limited scalability prevented statistical benchmarking on large datasets [...]

4. The authors discuss a conceptual improvement (line 114) potentially arising from the use of Normalizer. They should conduct ablation experiments to support this claim. Additionally, they could demonstrate how this improvement enhances cell-type-specific eQTL detection.

We thank Reviewer #2 for suggesting ablation experiments to further consolidate our statements. As noted above, CellRegMap does not have sufficient computational scalability for the ablation experiments. Therefore, we instead supported this claim through comparison with LogCPM normalization, a common baseline in scRNA-seq differential expression. We reported our new results in *Section 2.1 Comparison of single-cell normalization* in Supplementary Note and in Supplementary Figures 10 and 11:

Comparison of single-cell normalization:

We evaluated two normalization approaches within the airqtl framework: Normaliser (default) and log(CPM+10K) (namely LogCPM), assessing their impact on both standard and cell type-specific sceQTL mapping. LogCPM serves as a widely-used baseline method in scRNA-seq normalization studies. Our benchmarking utilized the same datasets and methodology as the airqtl-CellRegMap comparisons.

In standard sceQTL mapping, LogCPM exhibited elevated false positive rates (Supplementary Fig. 10ab) alongside expression-dependent underestimation bias of effect sizes (Supplementary Fig. 10cd), mirroring its previously documented limitations in scRNA-seq differential expression analyses [5].

The main text proposed that expression-dependent effect size biases could spuriously inflate cell type-specific signals, where mere expression-level changes might be misinterpreted as cell type-specific regulation. While CellRegMap's computational limitations prevented direct validation, airqtl's framework enabled testing this hypothesis by applying LogCPM normalization with identical acceleration benefits.

Consistent with our hypothesis, LogCPM demonstrated significantly higher false positive rates in cell type-specific sceQTL detection compared to Normaliser (Supplementary Fig. 11a). Furthermore, LogCPM underperformed Normaliser across all statistical benchmarks

(Supplementary Fig. 11bc), confirming the importance of proper normalization for accurate cell type-specific analyses.

Supplementary Figure 10: Superior statistical performance of Normaliser versus LogCPM in sceQTL mapping.

ab P-value distribution histograms (a) and QQ plots (b) for non-eQTLs show LogCPM's elevated false positive rate. Error bars in histograms were estimated as \sqrt{N} where N is the number of entries in each bin.

c Ground-truth (X) and estimated (Y) sceQTL effect sizes (dots) by LogCPM without stratification. Deviation of the best-fit linear model (dotted line) from the diagonal (dashed line) indicates the extent of overall effect size underestimation.

d Effect size estimation bias (top) and variance (bottom) for genes across expression quartiles (X).

Dashed line: perfect performance. Color: method.

Supplementary Figure 11: Normalisr outperforms LogCPM in cell type-specific sceQTL mapping.

a P-value distribution histograms for non-eQTLs (left) and non-specific eQTLs (right) without calibration show LogCPM's increased false positives. Error bars in histograms were estimated as $3\sqrt{N}$ where N is the number of entries in each bin.

b Ground-truth (X) and estimated (Y) cell type-specific sceQTL effect sizes (dots) by LogCPM without stratification. Deviation of the best-fit linear model (dotted line) from the diagonal (dashed line) indicates the extent of overall effect size underestimation.

c Effect size estimation bias (top) and variance (bottom) for genes across expression quartiles (X).

Dashed line: perfect performance. Color: method.

5. Overall, it is difficult to determine whether AIRQTL offers a computational speed improvement, a conceptual improvement, or both. The manuscript should clarify how AIRQTL compares with current state-of-the-art methods in these respects.

We thank Reviewer #2 for this question.

Airqtl first offers a conceptual improvement in AIR data structure and algorithms. AIR is first of its kind and is conceptual because it can accelerate other single-cell and spatial QTL studies or even existing sceQTL mapping methods, which we are investigating for a future paper. This conceptual improvement gives 1E8 computational speed improvement in sceQTL mapping. This unprecedented computational speed improvement further enables a series of conceptual improvements such as the first-ever objective method benchmarking and optimization for context specific sceQTL mapping, the first causal inference of cell state specific gene regulatory networks in primary cells, and dissection of drivers of cGRN heterogeneity.

To clarify this question from Reviewer #2 and several questions from Reviewers #1 and #3, we have revised major parts of this paper, such as “Airqtl overview” and Figure 1a. Alongside existing paragraphs in Introduction and Discussion, we believe the current version provides a better illustration of these conceptual and computational speed improvements.

As Reviewer #2 mentioned, we are very committed to comparing with current state-of-the-art methods. In this study, we compared airqtl against CellRegMap comprehensively and against FaST-LMM and GEMMA in computational complexities. We also have a long history of comparing against state-of-the-art methods in our previous method papers. However, a large portion of our results are novel and we are not aware of any existing method to compare with.

6. Line 206: Do the results imply that there are 641 proteins potentially acting as transcription factors (TFs), although their functions are currently unknown? The authors should investigate the potential functions of these proteins—e.g., whether they preferentially interact with known TFs or are associated with chromatin-related functions. Similarly, more insight into the functional roles of the proteins in the gene regulatory networks (GRNs) shown in Fig. 5d is needed.

We appreciate this biological question. The causal inference approach we utilized in this paper allows the unbiased identification of regulator genes irrespective of their mechanisms. In other words, the identified regulators do not have to be TFs. This is very different from gene regulations inferred from TF binding, where many binding events do not control expression level and can be regarded as false positives of gene regulation. It is also capable of detecting indirect effects A->C arising from A->B->C. This difference was discussed in our initial submission in Introduction, Discussion, and the sentence we mentioned 641 regulators.

To avoid confusing these 641 genes as TFs, we rephrased the sentence as (Line 211-212):

Notably, most regulators (641/681 or 94%) were not known TFs (based on known motifs [4], Methods) [...]

In our initial submission, we also investigated the functional roles of the proteins in the gene regulatory networks (GRNs) shown in Fig. 5d (old Lines 220-229, new Lines 227-236). i) We looked up and confirmed their known function in literature, as in references 43-53. ii) We performed GO enrichment analyses on their inferred target genes in Fig 5e. These two approaches provided consistent insights.

7. Regarding the GRN in Fig. 5, STAT1 is shown as a potential regulator. Since STAT1 is a TF, the authors are encouraged to investigate whether the predicted interactions are supported by TF binding data (e.g., ChIP-seq or TF binding motif analysis).

We thank Reviewer #2 for suggesting TF binding analysis. We searched the whole GEO for ChIP-seq experiments for STAT1 in human CD4+ T cells, manually examined each entry, and found one dataset. We restricted our search to human CD4+ T cells because this and previous studies have found high cell type specificity for GRNs.

We investigated and confirmed that the predicted interactions are supported by TF binding data. The new results are included in Lines 241-245 and Supplementary Figures 12 and 13, as quoted below:

We further found one STAT1 Chromatin Immunoprecipitation sequencing (ChIP-seq) dataset in human CD4+ T cells in the whole GEO [55]. Using this dataset, we identified

STAT1's putative direct targets through active binding sites (Methods). These exhibited substantial overlap with STAT1 targets in our cGRN (Supplementary Fig. 12, Supplementary Fig. 13), while the partial overlap confirmed airqtl's ability to capture indirect targets undetectable by ChIP.

Supplementary Figure 12: Concordance between airqtl-inferred and ChIP-seq-identified STAT1 target genes. Venn diagrams compare airqtl-inferred STAT1 targets (direct+indirect) with genes near STAT1 binding sites at varying genomic distances (panel title) in human CD4+ T cells, with (INFg) or without (Control) stimulation. The substantial but incomplete overlap confirms airqtl's ability to recover both direct ChIP-seq-detectable targets and indirect ChIP-seq-undetectable targets.

Supplementary Figure 13: High specificity of airqtl-inferred STAT1 target genes relative to ChIP-seq data. Target overlap metrics (Y) between airqtl predictions and ChIP-seq results are shown as: (top) gene counts and (bottom) percentage of predicted targets. Boxplots display baseline distributions for other master regulators against STAT1 ChIP-seq results, with orange dots marking STAT1-specific results. Analyses compare different genomic distances (X) and conditions (columns), revealing stronger agreement for proximal (v.s. distal) and INFg-stimulated (v.s. Control) targets --- consistent with known STAT1 biology and validating airqtl's cGRN inference accuracy. Combined: STAT1 targets found in either Control or INFg conditions. Whiskers indicate extrema.

8. There is a lack of detail on how the input data for AIRQTL was generated from the Randolph et al. dataset (line 383). This section should be expanded and clarified.

We appreciate Reviewer #2's request for additional methodological details regarding data preparation. The Randolph et al. dataset needed no additional processing other than reformatting. To clarify this, we added the following sentences (Lines 427-429) and uploaded our reformatting code onto Zenodo:

The Seurat object was provided in the original study and directly exported into the input files for airqtl without any processing. The code for reformatting is available on Zenodo under accession code 15925067.

9. We could run the software, but there is limited documentation and it mostly consists of scripts to re-run the experiments. This should be improved. Here are some additional comments.

We very appreciate Reviewer #2 for testing our software. We address each point below.

The Airqtl code is clean and well-structured, with good comments on the parameters.

We thank Reviewer #2 for the positive feedback.

There is no reference for the python version needed to run.

Python version needed was already provided in <https://github.com/grnlab/airqtl/blob/86c2e9872d0640ed5264d4059d86fd272b715939/pyproject.toml#L11> . The user's package manager such as pip automatically recognizes this parameter, seeks compatible solutions, and reports any inconsistency upon failure.

Data format can only be understood via a script. Author is encouraged to make use of Data Containers (AnnData / scanpy). This would make the useability of the method possible for a broader audience.

We have added commands ``airqtl utils convert_anndata`` and ``airqtl utils convert_vcf`` to convert AnnData or genotype data into the input format airqtl accepts (<https://github.com/grnlab/airqtl/blob/master/src/airqtl/pipeline/utils.py>). We have expanded the description of airqtl's input files (<https://github.com/grnlab/airqtl/tree/master/docs/tutorials/andolph>). We chose to use standard data formats such as txt (text) and tsv (tab-separated values) over specialized data containers because they are i) accepted by even broader audience across disciplines, ii) human readable without needing specialized software, and iii) more interoperable with other computational methods.

The method uses PCA reduced dimensions of the transcriptome data as covariates in the linear mixed model to control hidden structures. The package requires a covariates matrix as input, which can be obtained from the expression matrix.

Airqtl or this study does not use or require PCA reduced dimensions of the transcriptome data as covariates. However, airqtl does allow PCA reduced dimensions as a part of covariates. The user is free to compute and provide any covariate matrix as optional input, not restricted to PCA reduced dimensions. This is stated in old Lines 500-512 and new Lines 552-564.

There is no option/output to visualize cGRNs.

Cytoscape is an established, well-maintained and popular software to visualize biological networks. We used Cytoscape to visualize cGRNs in this study. Airqtl provides output files to be directly imported into Cytoscape for visualization. We believe the development of cGRN visualization functionality, especially better than Cytoscape, is beyond the scope of this study. On the other hand, visualization functionality suboptimal than Cytoscape presents no clear benefits to the research community.

Minor Points:

10. In Fig. 1, it is unclear how AIRQTL obtains cell-type-specific eQTL estimates. This should be clarified in the figure or its legend.

We thank Reviewer #2 for this suggestion. We have revised the legend to clarify:

Workflow of sceQTL mapping with airqtl for each selected cell type using only data from the corresponding cells.

11. Line 196 mentions chromatin rewiring. What data supports this claim? Was chromatin directly analyzed? If not, the authors should rephrase this statement to more accurately reflect the observed results (e.g., "rewiring of the GRN").

We thank Reviewer #2 for pointing out this potential confusion. We have rephrased the statement to (Lines 199-200):

[...] with differences confined to a small fraction (up to 3%), which could arise from chromatin-level regulation rewiring.

This revision more accurately acknowledges chromatin modification as one plausible explanation among several potential mechanisms underlying the observed GRN rewiring.

12. Line 259 should be rephrased. While eQTL analysis offers a robust statistical framework for linking genetic variants to expression traits, the manuscript does not provide or cite supporting evidence for this claim.

We thank Reviewer #2 for noting the lack of supporting evidence. We have added figure references and citations as supporting evidence for this claim (Lines 270-274):

Unlike GRNs based on DNA-binding or expression predictivity that struggle with perturbation outcome prediction [56], causal inference methods are specifically designed for this challenge with enormous successes across disciplines. This is evidenced by our accurate prediction of STAT1 perturbation effects (Fig. 5f) and previous successes with Normalizr [5].

Reviewer #2 (Remarks on code availability):

See main coments.

Reviewer #3 (Remarks to the Author):

This paper presents Airqtl, a novel computational method for mapping single-cell expression quantitative trait loci (sceQTLs) to infer cell state-specific causal gene regulatory networks (cGRNs). The motivation stems from the computational limitation of existing methods that are unable to scale to population-level scRNA-seq studies. By employing a compressed data structure called Array of Interleaved Repeats (AIR) together with GPU and other implementation optimization, airqtl significantly improves computational efficiency, achieving orders of magnitude acceleration compared to existing methods. The scalability of airqtl facilitates cell state-specific sceQTL mapping and allows robust and experimentally validated cGRNs. The author benchmarked airqtl against existing methods using both simulated and real datasets, and the real data analyses inferred cGRNs consistent with established biological knowledge.

This work makes contributions to sceQTL mapping and cGRN inference. However, I still have some major and minor comments listed below that may be helpful to improve the study.

We thank Reviewer #3 for their acknowledgement of our contributions.

Major comments:

1. Since computation acceleration is a major contribution of this work, the related computational complexity parts need further clarity and accuracy. In the 'SceQTL mapping — algorithmic accelerations' section, 1) for GRM estimation, $G(E)$ is a matrix of n_snp by n_cell , so the complexity of $G(E)$ multiplying its transpose should be $O(n_snp^2 * n_cell)$ instead of $O(n_snp * n_cell^2)$. The same issue occurred to GRM for G of size n_snp by n_donor . 2) In general, decomposition of matrix n by n requires a complexity of $O(n^3)$. Can the author elaborate on why GRM decomposition for the n_snp by n_snp matrix $G(E) * G(E)^T$ requires $O(n_cell^2 * n_donor)$ and ARI requires $O(n_donor^3)$? 3) The GRM transformation and SNP-gene correlation parts also need further clarification regarding the computational complexity.

We wish to thank Reviewer #3 for their careful review of our computational complexity analysis.

1) We have corrected the text to properly reflect that the genetic relationship matrix $K = G(E)^T * G(E)$ is n_cell by n_cell , with computation complexity $O(n_snp * n_cell^2)$ (Fig 1a and Line 347):

$$[...] K = G(E)^T * G(E) [...]$$

2) We elaborated on why GRM decomposition for the n_cell by n_cell matrix $G(E)^T * G(E)$ (not n_snp by n_snp matrix $G(E) * G(E)^T$, see 1) above) requires $O(n_cell^2 * n_donor)$ and ARI requires $O(n_donor^3)$ (Lines 352-357):

*Conventional methods decompose the GRM K in $\mathbb{R}^{n_cell * n_cell}$ in $O(n_cell^2 * n_donor)$ time, even after recognizing the GRM has a rank of n_donor due to row and column repeats. Because all cells from the same donor share the same germline genotypes, their corresponding rows and columns in $\text{mat } K$ would be identical. This allowed AIR to merge these duplicate rows and columns with rescaled variance, further reducing eigendecomposition or SVD time complexity $O(n_donor^3)$ (Supplementary Note).*

The technical details of how AIR accelerates SVD are described in Section 1.3.3 Singular value decomposition of Supplementary Note.

3) We have provided further technical clarification regarding the computational complexity for each part, including the GRM transformation and SNP-gene covariance in Section 1.4 Application in sceQTL mapping of Supplementary Note. This section details whether the input and output of each part is AIR, and how AIR can reduce the time complexity using our novel algorithms proposed in Section 1.3 Acceleration in matrix operations of Supplementary Note.

2. As computational complexity was discussed in the paper, fig 2 (a) and (b) are somewhat misleading, as they appear to suggest that the computational cost scales linearly with all dimension parameters. A brief discussion of the results on the practical and theoretical scalability should be added here.

We thank Reviewer #3 for this important observation about the scalability presentation. We fully agree that the computational cost does not scale linearly in general. Our initial submission utilized a log-log model to characterize running time (Eq 5). To eliminate this potential confusion, we explicitly mentioned the nonlinear scaling and extrapolation (not actual measurements) as (Lines 107-109):

Because running time often scales nonlinearly with problem dimensions, we employed a log-log model to extrapolate CellRegMap's running time for large-scale datasets and compared it with the actual runtime of airqtl (Methods).

3. In Fig. 1(a), if $E(R) = \sum_{in_donor}(E)$ denotes summation over all cells for a specific donor, is there information loss here (biologically or statistically), in comparison to using the cell-specific expression data? Can the author elaborate on this point?

We thank Reviewer #3 for this insightful question. We understand there is confusion regarding the difference between airqtl and pseudo-bulking approaches, as also mentioned by Reviewers #1 and #2. AIR differs fundamentally from pseudo-bulking in four key aspects:

i) Exact acceleration: AIR provides exact algorithmic acceleration, producing identical results to single-cell eQTL mapping that does not use AIR, while pseudo-bulking incurs information loss.

ii) Statistical power: AIR maintains the effective sample size as total cell count and preserves statistical power without information loss, unlike pseudo-bulking which reduces to donor count as demonstrated in SAIGE-QTL and other studies.

iii) Statistical modelling: AIR can account for cell level covariates and cell count imbalance between donors or contexts, which cannot be achieved by pseudo-bulking.

iv) Implementation: even in the SNP-gene covariance calculation, AIR performs log normalization on cell-level expression levels before aggregation, while pseudo-bulking aggregates expression read counts before log normalization, leading to different results.

We have revised the Results section to focus on how AIR accelerates several steps (Lines 64-72):

The core of airqtl is Array of Interleaved Repeats (AIR), a compressed data structure designed to efficiently handle repeated values, a pattern common in sceQTL mapping (Fig. 1a). SceQTL mapping requires genotype matrix expansion from donor- to cell-level,

leading to repeated values that incur substantial computational and storage overhead. AIR circumvents this inefficiency by storing only their original values and repeat patterns, enabling novel exact algorithms to accelerate basic matrix operations essential for sceQTL mapping, such as multiplication and singular value decomposition (Supplementary Note).

For instance, AIR factorizes cell-level genetic relationship matrix (GRM) in $O(n_{\text{donor}}^3)$ time complexity, compared to $O(n_{\text{donor}}n_{\text{cell}}^2)$ required by state-of-the-art methods like FaST-LMM [28] and GEMMA [29]. This principle underpins similar accelerations across all linear mixed model (LMM) steps in sceQTL mapping [...]

We also redrew Fig 1a to highlight how AIR algorithmically accelerates several other steps in sceQTL mapping:

Figure 1: Overview of algorithmic acceleration from AIR and sceQTL mapping with airqtl.

a AIR acceleration in key LMM steps. Existing methods expand the genotype matrix from donor-level (color) to cell-level, creating repeated rows and columns (dots connected by edges) that incur computational inefficiency. AIR instead operates on the compressed matrix and the number of repeats (2 3 1 shown), enabling mathematically equivalent but more efficient computations. Time complexity for each method (box color) is indicated below each step. [...] Box color (ab): method (yellow: existing; green: AIR/airqtl; gray: data). Matrix color (a) and schematic low-dimensional embedding color (b): donor. Each entry in the GRM K (a) can be shaded in up to two colors because its row and column indices can be associated with two different donors.

Also, in response to the feedback from all Reviewers, we improved the high-level and technical descriptions of the algorithmic accelerations and distinguished them from pseudo-bulking approaches in “Array of Interleaved Repeats (AIR)”, “SceQTL mapping — AIR accelerations” sections in Methods and Section 1.4 Application in sceQTL mapping in Supplementary Note.

4. In fig. 1, since (d) uses data from (c), the readability can be improved by connecting the two parts with shared notations. For instance, using C to denote the normalized covariates in both (c) and (d).

We thank Reviewer #3 for this suggestion. We have introduced shared notation for all panels in Fig. 1:

5. Comparing fig. 2(a) with 2(b), the second and third columns are a bit redundant as the numbers genes and donors do not change too much. Also, it would be good to clarify the fixed sized when one dimension size changes.

We appreciate Reviewer #3 for the suggestions regarding figure optimization.

While we acknowledge these columns might appear a bit redundant when present, we retained them for two major reasons. i) Their absence may raise doubts about airqtl's scalability along these two dimensions. ii) Complete representation of all parameter dimensions is necessary to properly characterize the scaling behavior and extrapolate running time through our log-log model (see Point 2 by Reviewer #3 above).

We have included the fixed sizes in the figure legends, which now read:

Fig 2ab: Running time (left Y) and computing cost (right Y) for sceQTL mapping on small-scale (a, defaults to $n_{SNP}=500, n_{gene}=500, n_{donor}=100, n_{cell}=1,000$) and realistic large-scale (b, defaults to $n_{SNP}=4,000,000, n_{gene}=5,000, n_{donor}=100, n_{cell}=50,000$) datasets with varying dimension sizes (X).

Fig 3ab: Running time (left Y) and computing cost (right Y) for cell type-specific sceQTL mapping on small-scale (a, defaults to $n_{SNP}=200, n_{gene}=1,000, n_{donor}=50, n_{cell}=2,000$) and large-scale (b, defaults to $n_{SNP}=200,000, n_{gene}=2,000, n_{donor}=100, n_{cell}=50,000$) datasets with varying dimensions. Color: method.

6. The accuracy of section 'SceQTL mapping — linear mixed model for cell state specificity' needs further improved or at least additional clarification. If A is a matrix of n_A by k , B is a matrix of n_B by k , the Kronecker product is of $n_A \times n_B$ by k^2 . The author also needs to clarify the self-product case.

We thank Reviewer #3 for identifying this accuracy issue. We have expanded this section by defining the operator as the modified row-wise Kronecker product (Lines 316-320):

Here we defined the *modified row-wise Kronecker product* \otimes satisfying the following conditions. i) For $A \in \mathbb{R}^{n_A \times k}$ and $B \in \mathbb{R}^{n_B \times k}$, $A \otimes B \in \mathbb{R}^{n_A n_B \times k}$ computes the element-wise product between each row pair of A and B . ii) Vectors are treated as 1-row matrices: $A \in \mathbb{R}^{n_A \times k}$ and $x \in \mathbb{R}^k$, $A \otimes x \in \mathbb{R}^{n_A \times k}$ regards x as a matrix in $\mathbb{R}^{(1 \times k)}$. iii) Self-product $A \otimes A \in \mathbb{R}^{n_A(n_A+1)/2 \times k}$ includes only unique row pairs: $A \otimes A = (A_1 \otimes A_1, \dots, A_1 \otimes A_{n_A}, A_2 \otimes A_2, \dots, A_2 \otimes A_{n_A}, \dots, A_{n_A} \otimes A_{n_A})$. iv) Note $\{0, 1\} \subset \{0, 1, 2\} \subset \mathbb{R}$.

7. Can the author elaborate on why an LMM with all linear and quadratic interaction terms was adopted? Was such quadratic interaction reflected in the result? How to interpret both statistically and biologically?

We appreciate Reviewer #3 for noticing our careful inclusion of interaction terms.

We included the linear terms because every of them is known to contribute to y , the gene expression level. Covariates such as library size (captured in C) are well-established confounders in scRNA-seq measurements of expression. Similarly, cell types or states (s) exhibit distinct expression profiles, and genotype (x) effects on expression (eQTLs) are extensively documented in prior literature.

We incorporated all quadratic interaction terms between variables with linear contributions. The Methods section has been expanded to clarify this choice (Lines 321-331):

The term $s \otimes x$ captures genotype-by-cell state interactions, with β quantifying the cell state-specific eQTL effect. We incorporated all other quadratic interaction terms as covariates, which offer several advantages for statistical inference. i) While confounders should be included to reduce false positives, the benefits of including non-confounder covariates, such as "prognostic covariates" that explain variance in the response variable, are less known. Prognostic covariates reduce residual variance and improve statistical power, even when not confounding $s \otimes x$ [57]. One example prognostic covariate is library size, which is known to confound scRNA-seq measurements of gene expression at experiment- and cell state-dependent levels. ii) Our comprehensive inclusion of quadratic terms follows established recommendations for covariate pre-selection (even if over-selection) before any statistical inference is drawn [58]. iii) On contrary, covariate selection based on association with target variable using the same dataset risks the statistical complexities of post-selection inference [59].

We wish to reiterate that our key contribution is computational acceleration that enables objective benchmarking and optimization of different statistical approaches. While airqtl matches or outperforms CellRegMap statistically, both airqtl and the broader sceQTL field have room for statistical improvement—a possibility unlocked by airqtl's speed. This was previously discussed (Lines 248–256, now Lines 259-268).

8. For equation (1), detailed description, including the dimension of each notation and their practical meaning, is needed to improve the readability of the manuscript.

We thank Reviewer #3 for suggestions for improving readability. We have expanded our description around Eq 1 in Lines 307-336, but omit quoting it here due to length and equation formatting.

Minor comments:

1. On page 2 line 20, there is a typo in 'hese approaches ...'.

We thank Reviewer #3 for noticing this. We have corrected this typo.

2. For better comparison, fig 2(c) and 2(d) should be combined with different colors denoting different methods.

We thank Reviewer #3 for this suggestion. We combined fig 2(c) and 2(d) with different colors denoting different methods for better comparison:

However, we observed that overlaying the data in a single plot obscured certain regions and made some points difficult to distinguish. For instance, teal-colored markers were nearly invisible near $X=0$ and $Y \neq 0$ due to the higher abundance of orange markers. Since this visualization risked misrepresenting the underlying distributions, we opted to retain the original separate plots to ensure accurate interpretation of the results.

We very appreciate all reviewers for their positive review. We have addressed the remaining reviewer comments in detail. Our response is marked in blue.

Reviewer #2 (Remarks to the Author):

The authors have clarified most of my requests and clarified the advances of Airqtl lies on the computational speed up, which allows the proposed analysis for the first time. They also explained (mostly technical reasons) why a broader benchmarking was not feasible. This resolves most of my requests. I do have a pending request regarding the stat1 chip-seq analysis.

Specific point

While the overlapp analysis (Sup. Fig. 12) suggests some relation between Stat1 targets and ChiP-seq targets; this analysis lacks statistical assesment. Authors could check the overlapp for randonly selected genes (same size as Stat1 targets) and estimate the enrichment with a hypergeometric test (or similar).

We have included hypergeometric test summary statistics for each panel of Sup. Fig. 12 in Supplementary Data 7 as requested.

Reviewer #3 (Remarks to the Author):

The author has addressed all my comments. I do not have any further comments.